# SEERATTENTION: LEARNING INTRINSIC SPARSE ATTENTION IN YOUR LLMS

## ABSTRACT

Attention is the cornerstone of modern Large Language Models (LLMs). Yet its quadratic complexity limits the efficiency and scalability of LLMs, especially for those with a long-context window. A promising approach addressing this limitation is to leverage the sparsity in attention. However, existing sparsity-based solutions predominantly rely on predefined patterns or heuristics to approximate sparsity. This practice falls short to fully capture the dynamic nature of attention sparsity in language-based tasks. This paper argues that attention sparsity should be *learned* rather than *predefined*. To this end, we design SeerAttention, a new Attention mechanism that augments the conventional attention with a learnable gate that adaptively selects significant blocks in an attention map and deems the rest blocks sparse. Such block-level sparsity effectively balances accuracy and speedup. To enable efficient learning of the gating network, we develop a customized FlashAttention implementation that extracts the block-level ground truth of attention map with minimum overhead. SeerAttention not only applies to post-training, but also excels in long-context fine-tuning. Our results show that at post-training stages, SeerAttention significantly outperforms state-of-the-art static or heuristic-based sparse attention methods, while also being more versatile and flexible to adapt to varying context lengths and sparsity ratios. When applied to long-context fine-tuning with YaRN, SeerAttention can achieve a remarkable 90% sparsity ratio at a 32k context length with minimal perplexity loss, offering a $5.67\times$ speedup over FlashAttention-2.

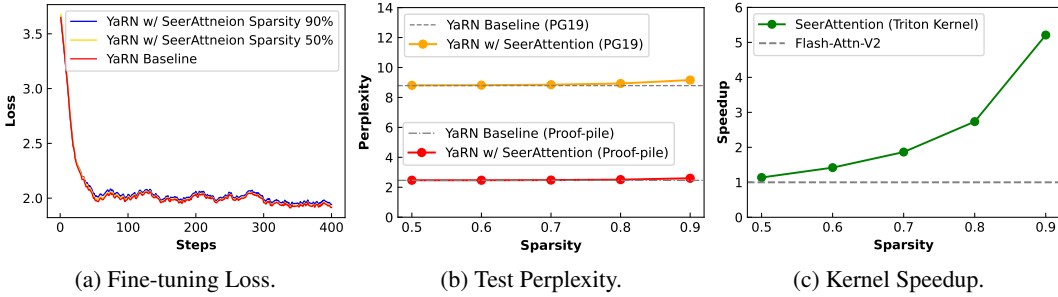

(a) Fine-tuning Loss.   (b) Test Perplexity.   (c) Kernel Speedup.

Figure 1: SeerAttention uses a learning-based approach to exploit attention sparsity of LLMs, applicable in both post-training and fine-tuning stages. By incorporating SeerAttention with YaRN (Peng et al., 2024) to extend a Llama-3-8B model from 8k to 32k context length, the loss curves for 50% to 90% sparsity are nearly identical to the dense YaRN baseline (a); For test perplexity, 50% sparsity achieves near-lossless performance, and even at 90% sparsity, the loss remains minimal (b); SeerAttention achieves up to 5.67x inference speedup at 90% sparsity over FlashAttention-2 (Dao, 2023);

## 1 INTRODUCTION

Attention is a fundamental mechanism in transformer-based LLMs (Vaswani, 2017). Despite its effectiveness, the quadratic complexity of attention requires substantial computation and memory resources, limiting the scalability and efficiency of LLMs, especially with a long-context window.

This problem has become an active research topic in the community. One potential solution is to replace the quadratic attention with cheaper architectures like linear attention or recurrent networks (Katharopoulos et al., 2020; Peng et al., 2023) with subquadratic complexity. These solutions, although efficient, struggle to match the efficacy of full attention, especially when the scale is large.

A promising approach with increasing interests is to leverage sparsity in attention. Sparsity commonly exists in attention maps, and it becomes more prominent in longer contexts. In certain LLM attention heads, the sparsity ratio can reach 95% or even 99%, posing great opportunities for efficiency improvement. However, prior studies often rely on predefined sparsity patterns or heuristics to approximate the attention mechanism (Jiang et al., 2024; Fu et al., 2024; Lee et al., 2024; Zhu et al., 2024; Han et al., 2023). The sparsity shown in attention maps varies significantly across different models, language inputs and attention heads, making predefined patterns or heuristics insufficient.

In this paper, we argue that attention sparsity should be *learned* rather than predefined. To achieve this, we introduce SeerAttention, a novel attention mechanism that enhances the standard attention with a learnable gate. During the forward pass of SeerAttention, the Q and K inputs are pooled and processed by the learnable gate to adaptively identify the important blocks, allowing the downstream block-sparse attention kernel to effectively reduce the I/O and computation by skipping unimportant blocks.

In the training of SeerAttention, the gate learns the block-wise attention sparsity from the model itself, i.e., using the attention map generated from the standard attention. However, FlashAttention (Dao et al., 2022), the state-of-the-art attention kernel used predominantly in LLMs, eliminates the explicit output of the intermediate attention maps via operation fusion to improve efficiency. This posts great challenges in our training processes, especially in long-context scenarios as the naïve manual attention implementation is slow and memory hungry. To address this challenge, SeerAttention customizes a FlashAttention kernel to extract our targeted block-wise attention map information without maintaining the original full size attention map. This new implementation achieves negligible overheads and significantly boosts the scalability of training process.

We evaluate SeerAttention in two settings: post-training, where only the gate parameters are learned using a small set of calibration data; and fine-tuning, where both the gate and weights of the original model are jointly optimized during long context extension. Our results demonstrate that SeerAttention surpasses state-of-the-art sparse attention methods like Minference (Jiang et al., 2024) and MoA (Fu et al., 2024). Notably, in contrast to previous methods that require careful calibration of sparse configuration for different settings, SeerAttention offers strong capabilities of adaptation to arbitrary context lengths and sparsity ratios. More importantly, the inherent learning capability of SeerAttention achieves near-lossless accuracy with 50% sparsity and minimal loss even with 90% sparsity during long-context fine-tuning (shown in Figure 1). The block-sparse kernel also demonstrates up to $5.67\times$ speedup over FlashAttention-2 dense baseline at 32k context size with 90% sparsity. Remarkably, with block-sparse pattern, SeerAttention exhibits the ability to learn more diverse patterns, including A-shape and Vertical-Slash, further demonstrating its versatility and performance.

Our contributions can be summarized as follows:

- We propose SeerAttention, an innovative attention mechanism that learns and leverages the intrinsic sparsity in attention to enhance efficiency for long-context LLMs.

- We develop a customized FlashAttention kernel that effectively obtains block-level attention map ground truth, enabling scalable learning of sparse attention.

- Experiments show that SeerAttention outperforms previous approaches in post-training, offers adaptability to various context lengths and sparsity ratios, and excels in long-context fine-tuning, maintaining near-lossless accuracy even at high sparsity levels.

## 2 BACKGROUND AND MOTIVATION

**Powerful but Complex Attention in Transformer.** The advent of attention mechanisms, particularly within the Transformer architecture (Vaswani, 2017), marked a significant advancement in natural language processing. Attention enables improved handling of long-range dependencies and

a better understanding of context by attending each token to every other token in the sequence, resulting in a quadratic memory and time complexity $O(n^2)$, where $n$ is the sequence length. This presents a significant challenge as the community moves towards LLMs that can process increasingly longer contexts. Many studies explore alternative attention mechanisms to mitigate this complexity. The Reformer architecture (Kitaev et al., 2020) reduces the complexity to $O(n \log n)$ and the linear attention mechanism (Katharopoulos et al., 2020) further decreases complexity to $O(n)$. Recently, there has been a trend of revisiting recurrent neural networks, leading to the proposal of new architectural frameworks such as RWKV (Peng et al., 2023), RetNet (Sun et al., 2023), and Mamba (Gu & Dao, 2023). Despite their promise of efficiency, these methods struggle to match the performance of full attention mechanisms, particularly with larger models and longer contexts.

**Intrinsic but Dynamic Sparsity in Attention.** Attention mechanisms inherently exhibit sparsity, which arises from the attention map $\mathbf{A}$ generated by $\mathbf{Q}$ and $\mathbf{K}$: $\mathbf{A} = \text{softmax}(\mathbf{Q}\mathbf{K}^\mathbf{T}/\sqrt{d})$. The softmax function often produces a multitude of negligible scores that can be treated as zeros without impacting model accuracy (Zaheer et al., 2020; Liu et al., 2021; Wang et al., 2021; Child et al., 2019). Attention sparsity becomes more pronounced with longer contexts, presenting opportunities to optimize inference speed. Unfortunately, this sparsity is dynamic, varying across different inputs and attention heads, each displaying distinct sparsity locations and ratios. Prior research has attempted to approximate attention sparsity using predefined patterns and heuristics (Fu et al., 2024; Jiang et al., 2024). Yet, these methods lack generality and often rely on handcrafted features, struggling to fully capture the sparsity behavior of attention mechanisms. The dynamic and input-dependent nature of attention sparsity echoes the principles of Mixture of Experts (MoE) models (Shazeer et al., 2017; Fedus et al., 2022) suggesting that sparsity should ideally be learned directly from data within the model itself. This approach would allow models to adaptively harness sparsity, improving efficiency while maintaining accuracy.

## 3 SEERATTENTION

SeerAttention adopts a fully learning-based approach to adaptively identify attention sparsity in LLMs and leverages the learned sparsity for efficient inference. To ensure efficiency on modern hardware like GPUs, we focus on learning block sparsity, which can seamlessly integrate with the tiling computation scheme of FlashAttention (Dao et al., 2022). Figure 2 illustrates the overall model architecture of SeerAttention, which augments conventional attention with a learnable gating module, termed *Attention Gate* (AttnGate). This module contains learnable parameters that identify the locations of significant blocks in the attention maps. By utilizing these block indices, the subsequent attention computation can employ a block-sparse FlashAttention kernel, significantly enhancing performance by reducing I/O and computation overhead.

### 3.1 ATTENTION GATE

The AttnGate module is designed to learn block-wise information with minimal overhead. It takes the original matrices $\mathbf{Q}$ and $\mathbf{K}$ as inputs and downsamples them using pooling along the sequence dimension. As shown in Figure 2a, for a given attention head, the sizes of the downsampled $\mathbf{Q}$ and $\mathbf{K}$ become $[seq/B, d]$, where $B$ is the block size. The downsampled $\mathbf{Q}$ and $\mathbf{K}$ are then processed through a linear layer and multiplied together, similar to the standard attention operation. This results in a matrix of size $[seq/B, seq/B]$, where each element corresponds to one block in the full attention map. With a typical block size of 64, the output of the AttnGate module is only $\frac{1}{4096}$ the size of the original attention map. During inference, by selecting the Top-k blocks in each row, the block-sparse FlashAttention kernel can efficiently load and process only the active blocks.

**Pooling Selection.** In SeerAttention, different pooling methods can be composed for the $\mathbf{Q}$ and $\mathbf{K}$ tensors, currently allowing for combinations of average, max, and min pooling. Multiple pooling operations can be applied to each matrix, with the resulting downsampled matrices concatenated before being fed into the linear layer. Experimental results indicate that the optimal combination is to use average pooling on $\mathbf{Q}$ and a combination of max and min pooling on $\mathbf{K}$ (details in Figure 10).

**Additional RoPE in Attention Gate.** Modern LLMs typically employ RoPE (Su et al., 2024) to encode positional information. If the AttnGate relies solely on the original RoPE in the model, i.e.,

feeding the AttnGate with $\mathbf{Q}$ and $\mathbf{K}$ after RoPE, the relative positional encoding properties will be lost because of the pooling operation. This compromises the AttnGate's ability to extrapolate to longer context lengths during training. Specifically, if the AttnGate is trained on 8k sequences, it struggles with inputs longer than 16k. To address this issue, we introduce a separate RoPE within the AttnGate. This RoPE can reuse the parameters from the original RoPE, but assigns position ids based on the starting positions of each block. This is equivalent to using a reduced rotational angle $\theta' = \theta/B$, but encode the position of each block.

## 3.2 BLOCK-SPARSE FLASHATTENTION INFERENCE KERNEL

Block sparsity is not officially supported in FlashAttention-2 (Dao, 2023), so we implement our own block-sparse FlashAttention kernel with Triton (Tillet et al., 2019) to speedup the inference of SeerAttention. It uses similar dataflow of FlashAttention-2 where $\mathbf{Q}$ is split across different warps. Each warp reads the sparse block indices generated by AttnGate and loads the corresponding $\mathbf{K}$ and $\mathbf{V}$ blocks on-chip for computation. This approach efficiently reduces both I/O and computation overhead by skipping non-activated blocks.

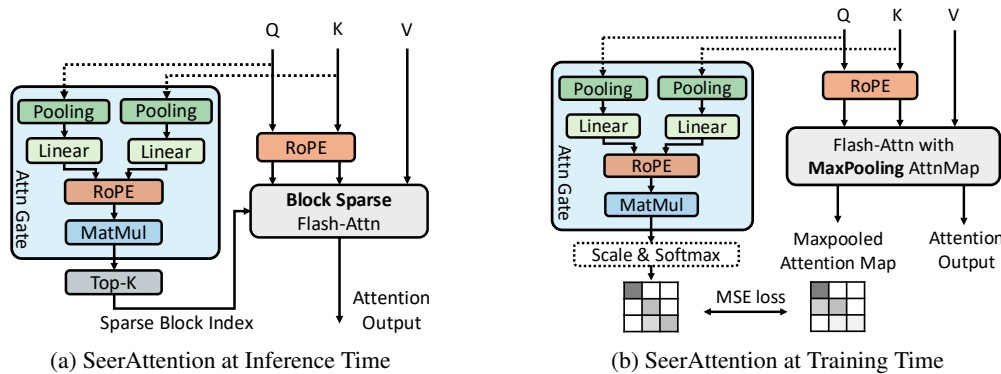

(a) SeerAttention at Inference Time      (b) SeerAttention at Training Time

Figure 2: SeerAttention Architecture. (a) SeerAttention incorporates an efficient module, AttnGate, to adaptivily identify sparse block locations in attention maps. (b) During training, SeerAttention uses the max-pooled attention map of full attention as ground truth to guide the AttnGate.

## 4 TRAINING SEERATTENTION

While the introduced SeerAttention architecture is straightforward, training it is challenging. Jointly training the gate and attention from scratch, as in MoE, is costly and difficult. Fortunately, unlike MoE, where gating network must learn expert selection from scratch, the AttnGate in SeerAttention has a ground truth in standard attention as guidance.

### 4.1 TRAINING THE ATTENTION GATE

We train the AttnGate to learn block-level sparsity by using the 2D max-pooled attention map from full attention as ground truth, as illustrated in Figure 2b. To align distributions, the AttnGate's output is scaled and passed through a softmax, similar to standard attention mechanisms. Additionally, the max-pooled attention map is row-normalized to sum to 1, consistent with the softmax output. Mean-Square-Error (MSE) loss is used in training. This auto-regressive training scheme also enables flexible usage of SeerAttention, allowing users to adjust the Top-k ratio to balance accuracy and efficiency with a single model.

### 4.2 FLASHATTENION WITH MAX-POOLING: A CUSTOMIZED TRAINING KERNEL

Obtaining the max-pooled attention map for training is non-trivial especially in long-context scenarios. Modern LLMs rely on FlashAttention, which fuses operations and doesn't explicitly compute the attention map. The naïve mannul implementation is impractical due to quadratic memory complexity. To address this challenge, we customize an efficient kernel that directly outputs the

max-pooled attention map by modifying FlashAttention but largely reuses its original computation flow. Figure 3 shows the pseudo code and diagram of this customized kernel.

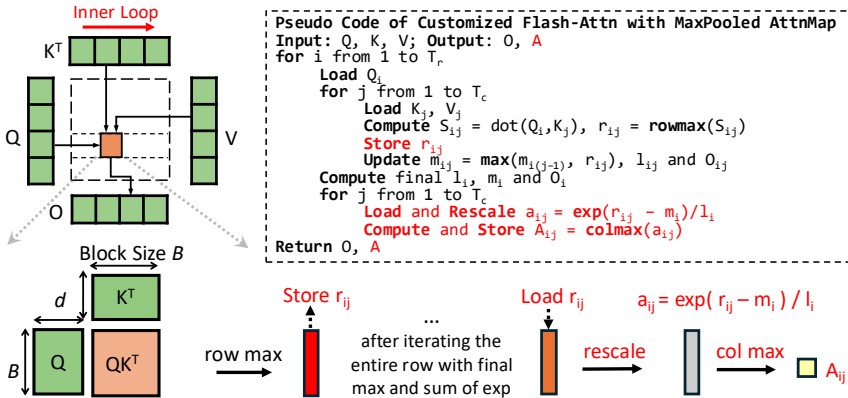

Figure 3: Efficient FlashAttention kernel with pooling of attention map.

Normally, the softmax function ensures numerical stability by subtracting the maximum value before applying the exponential operation. FlashAttention computes the local row max of each block, and gradually updates the global maximum through iteration:

$$S_{ij} = Q_i K_j^T; \ r_{ij} = \mathrm{rowmax}(S_{ij}); \ m_{ij} = \max(m_{i(j-1)}, r_{ij}) \tag{1}$$

where $r_{ij}$ is typically treated as a temporary result. However, we store it in HBM and rescale it later with the final global max $m_i$ and sum of exp $l_i$ after the iteration:

$$a_{ij} = \exp(r_{ij} - m_i)/l_i \tag{2}$$

This $a_{ij}$ represents the correct row max of the original attention block. With that, 2D max-pooling is achieved by applying a column max over $a_{ij}$. This introduces only minor overhead (storing and rescaling $r_{ij}$) but significantly improves the efficiency of obtaining the ground truth. Detailed code is available in the Appendix A, and the overhead analysis is in Figure 8.

### 4.3 APPLY SEERATTENTION IN POST-TRAINING AND FINE-TUNING STAGES

**Post-training.** SeerAttention can be directly applied to a pre-trained model. In this case, only the weights of AttnGate are learned and updated, leaving the original model weights unchanged. This method is highly efficient and cost-effective, requiring gradients solely for the AttnGate, and quickly converges using minimal calibration data. The learned gate also allows for adjustable Top-k ratios during inference, providing a flexible tradeoff between accuracy and efficiency.

**Fine-tuning.** SeerAttention can also be applied to long-context extension fine-tuning, enabling improved model performance and higher sparsity ratios. In practice, to ensure stable training, the AttnGate is first initialized using the post-training method before fine-tuning the entire model. During fine-tuning, we fix the Top-k ratio and use both the original training loss and the attention map MSE loss.

## 5 EXPERIMENTS

In this section, we evaluate both the accuracy and efficiency of SeerAttention. The accuracy is evaluated under two distinct scenarios: (1) post-training stage and (2) long-context extension fine-tuning stage. For the efficiency evaluation, we present kernel-level and end-to-end latency speedup results across various sparse configurations. In our current experiments, block-size $B$ for the model and kernel is fixed at 64 and AttnGate currently solely applies in the prefill stage.

**Models, Tasks and Baselines.** We apply SeerAttention to the pre-trained models Llama-3.1-8B (Dubey et al., 2024) and Mistral-7B-v0.3 (Jiang et al., 2023) to assess its impact on LLM perplexity across different AttnGate designs and sparsity configurations. For perplexity evaluation, we

use the PG19 (Rae et al., 2019) and Proof-pile (Azerbayev et al.) test splits. Following YaRN (Peng et al., 2024), 10 documents over 128k tokens are sampled from Proof-pile, while all documents exceeding 128k tokens from PG19 are selected. The input sequences are truncated to evaluation context length before feeding into the model. We also conduct experiments using an instruction-tuned model, Llama-3.1-8B-Instruct, and compare SeerAttention with two state-of-the-art sparse attention methods, MoA (Fu et al., 2024) and MInference (Jiang et al., 2024), on the LongBench (Bai et al., 2023) benchmark, perplexity and efficiency. MoA uses an offline search scheme to apply static sparse patterns across different attention heads, while MInference dynamically generates sparse indices using heuristic methods for each head based on pre-defined sparse patterns.

**Post-training Setup.** We use the RedPajama (Computer, 2023) dataset for calibration, chunked into 64k and 32k segments for Llama-3.1 and Mistral, respectively.We use a learning rate of 1e-3 with cosine decay and the global batch size of 16. The AttnGate is only trained in 500 steps using the ground truths from our customized FlashAttention kernel, with DeepSpeed (Rasley et al., 2020) stage 2 optimization on 4 A100 GPUs. As only AttnGate parameters are learned and updated in post-training, this process can be completed with hours.

**Long-context Extension Fine-tuning Setup.** We extend the context size of a Llama-3-8B model from 8K to 32K, following the setup from YaRN (Peng et al., 2024), while introducing attention sparsity via SeerAttention. The Top-k number in the AttnGate is fixed during the forward pass to allow the model to adapt to the sparsity. We use a learning rate of 1e-5 with linear decay and a global batch size of 8 on RedPajama dataset. The entire model weights are fine-tuned on 4 A100 GPUs with DeepSpeed stage 3 optimization.

## 5.1 ACCURACY OF POST-TRAINING

**Perplexity on Pre-trained Models.** Figure 4 shows the perplexity results on the Proof-pile dataset for both Llama-3.1-8B and Mistral-7B-v0.3 across different context lengths and sparsity ratios. It should be noted that the results for each model come from the same checkpoint with trained AttnGates, and different sparsity ratios are achieved by adjusting the value of $k$ in the Top-k. The results show that SeerAttention only slightly increases perplexity as the sparsity ratio increases, compared to full attention. For instance, with the Mistral-7B model at a 32k context size, SeerAttention achieves a perplexity of 2.45, compared to the baseline of 2.29, despite introducing a significant 90% attention sparsity. Figure 4 also demonstrates that longer context lengths allow for greater sparsity with minimal accuracy degradation.

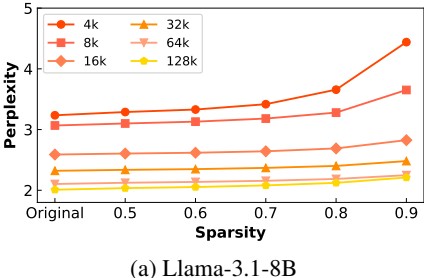
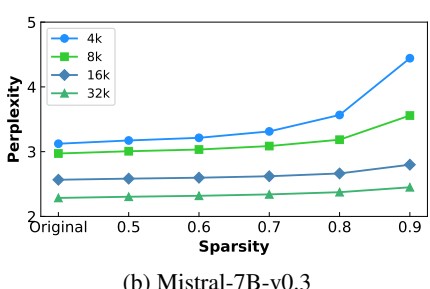

(a) Llama-3.1-8B                    (b) Mistral-7B-v0.3

Figure 4: Perplexity results on Proof-pile across various context lengths and sparsity ratios. Note that results on various sparsity ratios comes from the same trained AttnGates by only adjusting the Top-k ratios. Longer context sizes allow for higher sparsity with minimal performance loss.

**Perplexity Comparison with Related Works.** Table 1 compares the perplexity of SeerAttention at post-training with MoA and MInference, using the Llama-3.1-8B-Instruct model on the PG19 dataset. For MoA, we adopt their "KV Sparsity" in 0.5 which means "Attention Sparsity" in 0.35. For MInference, we use their official setup, where all attention heads choose the "Vertical-Slash" sparsity pattern for Llama-3.1-8B-Instruct. Since MInference dynamically generates sparse indices for each input, we record their **average** attention sparsity across different context lengths for comparison. SeerAttention outperforms MoA and MInference even with higher sparisty in most cases,

except at the 128k context length. This is likely due to MInference applies varying sparsity per head, whereas the fixed sparsity ratio across all heads in SeerAttention. Varying sparsity per head could be applied to SeerAttention for enhancement, which remains a topic for future work.

Table 1: Comparing the perplexity of SeerAttention at post-training with MoA and MInference, using the Llama-3.1-8B-Instruct model on the PG19 dataset.

| | Sparsity $s$ | Evaluation Context Length | | | | |
| | | 8k | 16k | 32k | 64k | 128k |
|---|---|---|---|---|---|---|
| Original | 0.0 | 10.03 | 9.88 | 9.92 | 9.97 | 10.03 |
| MoA | 0.35 | 10.07 | 9.97 | 10.02 | 10.13 | OOM |
| MInference | | 10.12 | 10.06 | 10.24 | 10.43 | **10.89** |
| | | $s = 0.37$ | $s = 0.55$ | $s = 0.69$ | $s = 0.80$ | $s = 0.9$ |
| SeerAttention | 0.4 | **10.06** | 9.92 | 9.96 | 10.10 | 10.29 |
| | 0.5 | 10.08 | 9.94 | 9.99 | 10.15 | 10.38 |
| | 0.6 | 10.12 | **9.96** | 10.04 | 10.21 | 10.50 |
| | 0.7 | 10.18 | 10.01 | **10.10** | 10.29 | 10.71 |
| | 0.8 | 10.30 | 10.07 | 10.18 | **10.39** | 11.18 |
| | 0.9 | 10.75 | 10.24 | 10.30 | 10.56 | 13.20 |

**LongBench Evaluation.** To evaluate performance on instruction-following tasks, we conduct experiments on LongBench, a long-context understanding benchmark, and compare the results with MoA and MInference using the Llama-3.1-8B-Instruct model. As shown in Table 2, SeerAttention consistently outperforms both MoA and MInference under similar or higher sparsity ratios.

Table 2: Comparing the accuracy of SeerAttention at post-training with MoA and MInference on LongBench.

| Model | Attention | Sparsity $s$ | LongBench | | |
| | | | 0-4k | 4-8k | 8k+ |
|---|---|---|---|---|---|
| | Original | 0.0 | 55.32 | 53.98 | 52.90 |
| | MoA | 0.35 | 50.74 | 49.84 | 51.89 |
| Llama-3.1-8B-Instruct | MInference | | 55.23 | 53.87 | 52.18 |
| | | | $s = 0.06$ | $s = 0.25$ | $s = 0.45$ |
| | SeerAttention | 0.1 | **55.91** | 54.32 | 53.28 |
| | | 0.25 | 55.00 | **54.09** | 52.22 |
| | | 0.5 | 52.40 | 52.85 | **52.43** |

## 5.2 ACCURACY OF LONG-CONTEXT EXTENSION FINE-TUNING

We follow YaRN (Peng et al., 2024) to extend the context size of a Llama-3-8B model from 8k to 32k. We integrate SeerAttention into YaRN and compare the performance against the YaRN dense baseline and the post-training SeerAttention applied after YaRN. Figure 1a presents the loss curves of the YaRN dense baseline and SeerAttention at 50% and 90% sparsity. The curve at 50% sparsity nearly overlaps with the baseline, while the curve at 90% sparsity shows slightly higher loss. Table 3 displays the test perplexity on the PG19 and ProofPile datasets evaluated at a 32k context length. The YaRN dense baseline achieves perplexity scores of 8.79 and 2.46, respectively. Post-training SeerAttention results in increased perplexity. When applying SeerAttention during the YaRN extension fine-tuning, it maintains near-lossless performance at 50% sparsity (with scores of 8.81 and 2.47), and even at 90% sparsity, the loss remains minimal.

Table 3: Perplexity of YaRN baseline, SeerAttention after YaRN and YaRN with SeerAttention.

| | YaRN | Post-training SeerAttention after YaRN | | | | | YaRN with SeerAttention | | | | |
| Sparsity | 0.0 | 0.5 | 0.6 | 0.7 | 0.8 | 0.9 | 0.5 | 0.6 | 0.7 | 0.8 | 0.9 |
|---|---|---|---|---|---|---|---|---|---|---|---|
| PG19 | 8.79 | 9.16 | 9.30 | 9.48 | 9.73 | 10.18 | 8.81 | 8.82 | 8.85 | 8.93 | 9.16 |
| Proof-pile | 2.46 | 2.53 | 2.57 | 2.61 | 2.68 | 2.85 | 2.47 | 2.47 | 2.48 | 2.51 | 2.60 |

## 5.3 EFFICIENCY EVALUATION

We evaluate the efficiency of SeerAttention using our implementation of Triton (Tillet et al., 2019) kernels. We evaluate the kernel-level as well as end-to-end speedup using a Llama-3.1-8B-Instruct on a single A100 GPU. Results are compared to FlashAttention-2 (dense baseline), MoA and MInference.

### 5.3.1 KERNEL EVALUATION

**Negligible Overhead in AttnGate and Top-k.** Figure 5 shows the kernel-level latency breakdown of SeerAttention. It demonstrates that the overhead introduced by the AttnGate and Top-k operations during inference is minimal. For instance, at a context length of 32k and a sparsity of 0.5, the AttnGate and Top-k contribute only 1% and 2% to the total latency, respectively. In the cases of 128k sequence length, the relative overhead almost diminishes.

**Block-Sparse FlashAttention Kernel Speedup.** Figure 5 also shows that our kernel demonstrates linear speedup over various sparsity levels. At a sequence length of 128k with 90% sparsity, SeerAttention achieves a speedup of 5.47× compared with FlashAttention-2 on a single A100 GPU. While the current implementation is based on Triton, further performance gains are possible by optimizing the kernel using CUDA in future work.

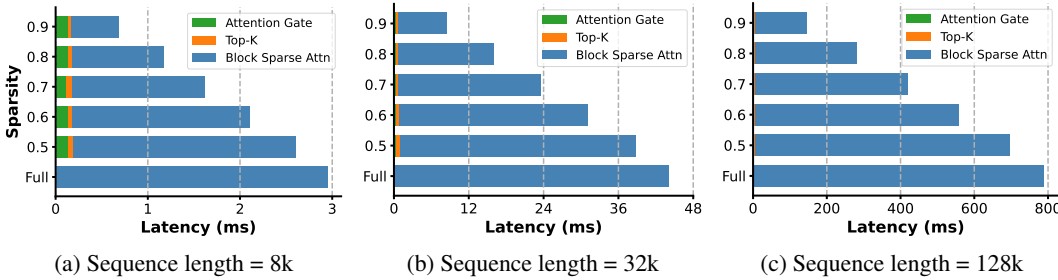

(a) Sequence length = 8k      (b) Sequence length = 32k      (c) Sequence length = 128k

Figure 5: SeerAttention time breakdown compared to FlashAttention-2. At sequence length 128k with 90% sparity ratio, SeerAttention speeds up attention computation by 5.47× over FlashAttention-2.

**Compared to Related Works.** We compare the speedup of SeerAttention with MoA and MInference. MInference uses offline calibration to identify a pre-defined sparse pattern for each layer. For Llama-3.1-8B-Instruct model, MInference consistently uses "Vertical-slash" pattern across all layers. During runtime, MInference will dynamically generate non-zero indices based-on their approximation algorithm. On the other hand, MoA uses "A-shape" blocks as their sparse pattern and calibrate the shape parameters offline under given sparsity constraint.

Figure 6 shows the sparsity v.s. speedup plots of different methods on 8k, 32k, 128k sequences length, where the speedup baseline is FlashAttention-2. The sparsity statistics were collected on PG19 datasets. For MoA, we generated the sparse configurations under their 0.5 overall "KV-sparsity" constraints, which corresponds to an average of 0.35 sparsity in attention. The results demonstrates that SeerAttention outperforms both MoA and MInference in most cases. At 128k, the performance of all three methods converges, where the benefits of sparsity significantly outweigh the associated overhead.

### 5.3.2 END-TO-END SPEEDUP

To assess the end-to-end speedup of our method, we measured the average prefilling time, or time-to-first-token (TTFT), using the Llama-3.1-8B-Instruct model. Following the experimental setup used in MoA, we also recorded the average sparsity statistics for each method. The results show that SeerAttention consistently achieves lower latency compared to MInference, even with lower sparsity ratios. As for MoA, it requires an exhaustive search for different sparse configurations under varying sparsity constraints, which is time-consuming. Therefore, we only compared against its default configuration.

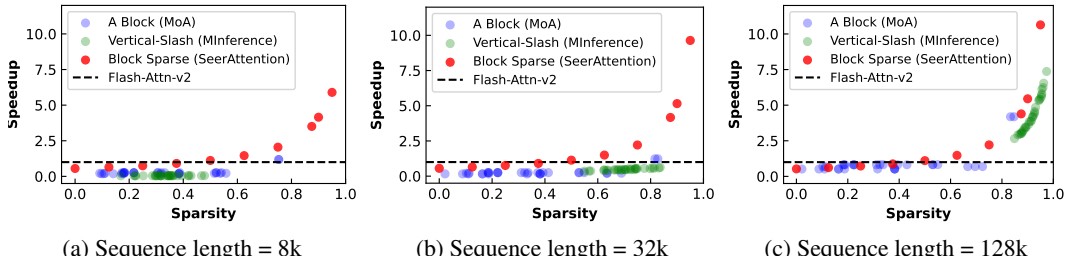

(a) Sequence length = 8k    (b) Sequence length = 32k    (c) Sequence length = 128k

Figure 6: SeerAttention block sparse FlashAttention inference kernel speedup.

Table 4: Time to First Token results (s).

| Latency (Sparsity) | Evaluation Context Length | | | | |
|---|---|---|---|---|---|
| | 8k | 16k | 32k | 64k | 128k |
| FlashAttn-2 | 0.90 (0) | 1.95 (0) | 4.63 (0) | 10.09 (0) | 35.54 (0) |
| MoA | 1.29 (0.35) | 3.44 (0.35) | 10.34 (0.35) | 36.34 (0.35) | OOM |
| MInference | 2.33 (0.37) | 3.10 (0.65) | 4.68 (0.77) | 8.21 (0.86) | 14.38 (0.95) |
| SeerAttention | 0.78 (0.50) | 1.65 (0.60) | 3.60 (0.70) | 7.69 (0.80) | 13.37 (0.95) |

## 6 ANALYSIS AND ABLATION

**Visualization of Learned Attention Maps.**  The AttnGate module automatically learns diverse sparse patterns without any prior knowledge or heuristics. Figure 7 shows several example outputs from AttnGate, including (a) "A-shape," (b) "Vertical," (c) "Slash" with empty vertical spaces, (d) block sparsity along the diagonal, and (e) random patterns. These patterns not only encompass but also extend beyond those observed in previous works such as MoA and MInference, showcasing the versatillty of our learning based methods.

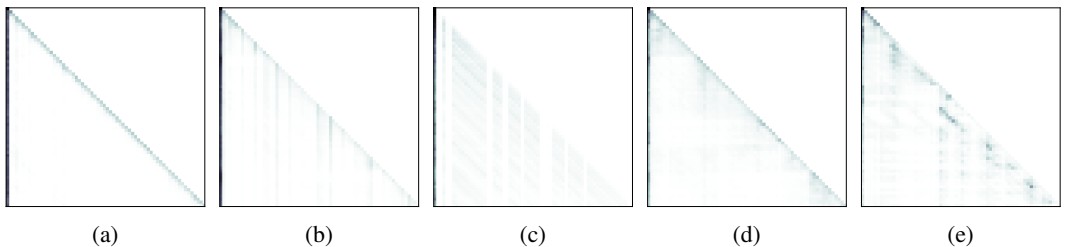

(a)                (b)                (c)                (d)                (e)

Figure 7: Visualization of the AttnGate's outputs.

**Analysis of FlashAttention with Max-Pooling Training Kernel.**  We evaluate our customized FlashAttention kernel with maxpooling attention map for scalable training of SeerAttention by comparing against with PyTorch naïve manual attention implementation and FlashAttention-2. As shown in Figure 8b, the PyTorch kernel runs out of memory (OOM) when the sequence length exceeds 4k, while our customized kernel costs similar peak memory usage compared to FlashAttention-2. Regarding latency, since PyTorch encounters OOM for sequences longer than 8K, the attention operations per head into a loop to assess kernel-level latency. Figure 8b shows that the latency overhead introduced by the additional pooling operation is minimal compared to the FlashAttention-2, while the PyTorch implementation suffers from a significant slowdown.

**RoPE Ablation.**  In our experiments, we found that incorporating an additional RoPE module in the AttnGates, shown in Figure 2, significantly enhances its ability to extrapolate context length when training AttnGates. Figure 9 shows the results with and without RoPE in AttnGate on PG19 with Llama-3.1-8B model using 8k length training data. The AttnGate with RoPE shows very consistent performances on larger context lengths despite only trained with 8k length data.

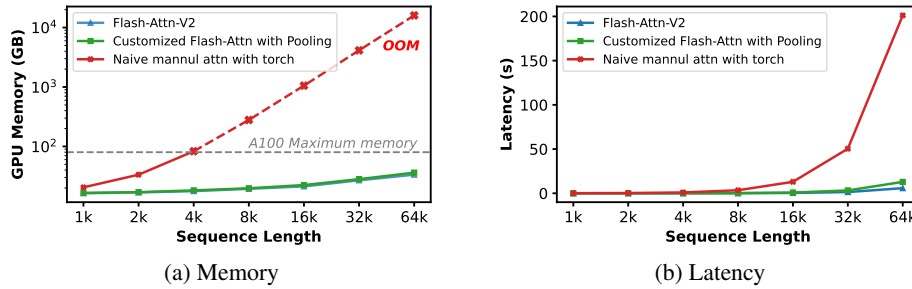

(a) Memory  (b) Latency

Figure 8: Memory and latency of customized FlashAttention with max-pooling training kernel.

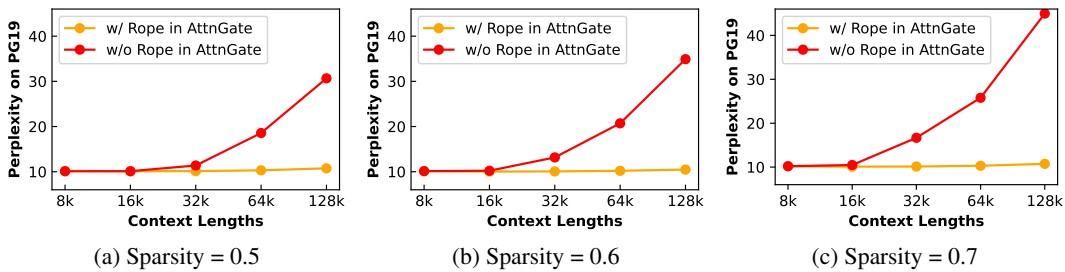

(a) Sparsity = 0.5  (b) Sparsity = 0.6  (c) Sparsity = 0.7

Figure 9: Perplexity with and without RoPE in AttnGate.

**Pooling Ablation.** Different pooling methods can be composed in the AttnGates. To study the best configuration, we test all the possible combinations of pooling in $\mathbf{Q}$ and $\mathbf{K}$ choosing from average, max and min pooling. There are in total 49 combinations. We train each configuration in post-training setting using Llama-3.1-8B model with 32k length data and test the perplexity on PG19 dataset with 128k evaluation context length. Figure 10 shows the Top-12 configurations. The one with average pooling on $\mathbf{Q}$ and max plus minpooling on $\mathbf{K}$ performs the best. This may relate to the observations in LLM quantization that $\mathbf{K}$ often exhibits more outliers.

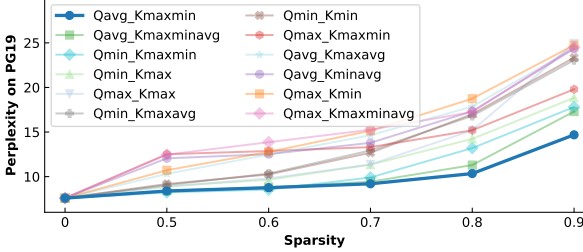

Figure 10: Perplexity of SeerAttention with different pooling methods.

## 7 CONCLUSION AND FUTURE WORK

This paper presents SeerAttention, a new attention mechanism that learns and leverages the intrinsic sparsity in attention to boost long-context LLMs. Our experiments demonstrate that SeerAttention not only outperforms previous approaches in post-training scenarios, but also excels in long-context fine-tuning, maintaining near-lossless accuracy even at high sparsity levels. For future work, there are several promising directions to explore for improving and expanding the capabilities of SeerAttention. One key area is enhancing the training methodologies for SeerAttention, such as applying SeerAttention in long-context continued pre-training with more training tokens to achieve higher sparsity without sacrificing accuracy. Another important avenue is applying SeerAttention in the decoding stage of LLMs. While this work primarily focuses on the prefill phase, it remains an open question whether the learned attention sparsity can similarly benefit the efficiency and accuracy of the decoding process.

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

# A  APPENDIX

---

**Algorithm 1:** Customized FlashAttention with Max-pooling Kernel

---

**Input:** Matrices $Q, K, V \in \mathbb{R}^{N \times d}$ in HBM, block sizes $B_c, B_r$
**Output:** Output $O$, logsumexp $L$ and attention map $A$

1  Divide $Q$ into $T_r = \left\lceil \frac{N}{B_r} \right\rceil$ blocks $Q_1, \ldots, Q_{T_r}$, of size $B_r \times d$ each
2  Divide $K, V$ into $T_c = \left\lceil \frac{N}{B_c} \right\rceil$ blocks $K_1, \ldots, K_{T_c}$ and $V_1, \ldots, V_{T_c}$, of size $B_c \times d$ each
3  Divide the output $O \in \mathbb{R}^{N \times d}$ into $T_r$ blocks $O_1, \ldots, O_{T_r}$, of size $B_r \times d$ each
4  Divide the logsumexp $L$ into $T_r$ blocks $L_1, \ldots, L_{T_r}$, of size $B_r$ each
5  Divide attention score $A \in \mathbb{R}^{T_r \times T_c}$ into $(T_r \times T_c)$ blocks $A_0^{(0)}, \ldots, A_{T_r}^{(T_c)}$, initialize $A_i^{(j)} = (0)_{1 \times 1}$
6  **for** $i = 1$ **to** $T_r$ **do**
7      Load $Q_i$ from HBM to on-chip SRAM
8      On chip, initialize $O_i^{(0)} = (0)_{B_r \times d}, \ell_i^{(0)} = (0)_{B_r}, m_i = (-\infty)_{B_r} \; r_i = (-\infty)_{B_r}$
9      **for** $j = 1$ **to** $T_c$ **do**
10          Load $K_j, V_j$ from HBM to on-chip SRAM
11          On chip, compute $S_i^{(j)} = Q_i K_j^T \in \mathbb{R}^{B_r \times B_c}$
12          On chip, compute $m_i^{(j)} = \max(m_i^{(j-1)}, \text{rowmax}(S_i^{(j)}))$
13          On chip, compute $\hat{P}_i^{(j)} = \exp(S_i^{(j)} - m_i^{(j)})$
14          Update $\ell_i^{(j)} = \ell_i^{(j-1)} + \text{rowsum}(\hat{P}_i^{(j)})$
15          On chip, compute $O_i^{(j)} = \text{diag}(\exp(m_i^{(j-1)} - m_i^{(j)}))^{-1} O_i^{(j-1)} + \hat{P}_i^{(j)} V_j$
16          Store $r_i^{(j)} = \text{rowmax}(S_i^{(j)})$
17      **for** $j = 1$ **to** $T_c$ **do**
18          Update $r_i^{(j)} = \text{diag}(\ell_i^{(T_c)})^{-1} \exp(r_i^{(j)} - m_i^{(T_c)})$
19          On chip, compute $D_i^{(j)} = \text{colmax}(r_i^{(j)})$
20          Write $A_i^{(j)}$ to HBM as $(i, j)$-th block of $A$
21      On chip, compute $O_i = \text{diag}(\ell_i^{(T_c)})^{-1} O_i^{(T_c)}$
22      On chip, compute $L_i = m_i^{(T_c)} + \log(\ell_i^{(T_c)})$
23      Write $O_i$ to HBM as the $i$-th block of $O$
24      Write $L_i$ to HBM as the $i$-th block of $L$
25  **return** $O$, $L$, $A$

---

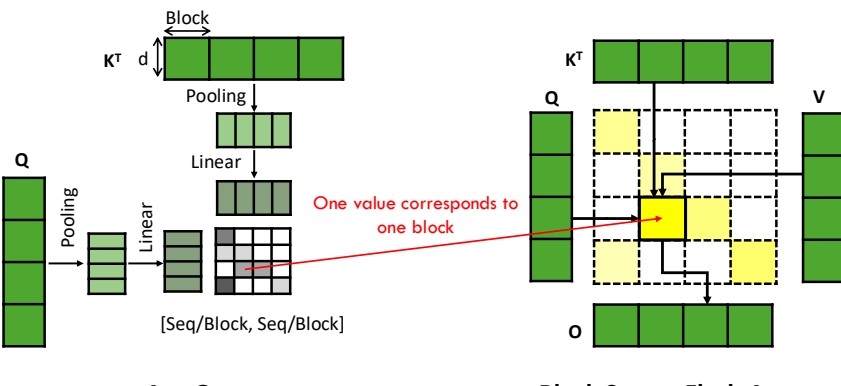

Figure 11: Dataflow of AttnGate. The AttnGate downsamples Q and K in seq dimention and further process with linear layers. Each value in the output tensor of AttnGate correspond to one block in FlashAttention computation. By selecting the TopK blocks with highest score, a block sparse Flash-Attn kernel can skip the computation of non-selected blocks.

