# OpenReview forum: "SeerAttention: Learning Intrinsic Sparse Attention in Your LLMs"
_ICLR.cc/2025/Conference — Submitted to ICLR 2025_

### Official Review · Reviewer_6XFX · 2024-10-29

**Soundness:** 3
**Presentation:** 4
**Contribution:** 3
**Rating:** 5
**Confidence:** 5

**Summary:**

SeerAttention proposes an efficient attention implementation that reduces attention computation. It uses the learned attention-gating network, which is the same attention mechanism but with a pooled sequence.

**Strengths:**

1. Simple yet effective.

The overall paper is crystal clear and intuitive. I have no concerns about the presentation and methodology explanation.

**Weaknesses:**

1. Require of training.

Even this paper claims that sparsity should be learned, but in most cases, it is quite expensive to do so with a large-scale language model such as llama3 70B or 405B.
I think the requirement of learning sparsity patterns is the extreme downside of this methodology, and this direction has also been tried in the past years and always not chosen in practice scenarios due to required training and additional engineering (TDSA: https://arxiv.org/abs/2110.11299, Reformer: https://arxiv.org/abs/2001.04451)

I have concerns that the efficiency of this approach is really better than MInference with training free context extension methods (Self-Extend) during the whole life cycle of LLM. I think if the method should be trained every time a new LoRA or new model is coming, its effectiveness will be quite limited and will have the same limitations as previous methods (TDSA, Reformer, Mamba: https://arxiv.org/abs/2312.00752, etc. (training requires methods).) Therefore, I suggest you add how the trained SeerAttention can adapt to new tasks or unseen weights (LoRA).

Furthermore, is there really no way to pool tokens and approximate attention patterns without training or minimal training (a few steps)? I think there should be some great way to do something like QUEST (https://arxiv.org/abs/2406.10774). I strongly suggest authors try to investigate or at least compare with simple heuristics (averaging, absolute maximum, QUEST-style reducing)

Additionally, the requirement for training may lead to bad generalizability, as shown by the perplexity exploding in Table 1's 128k context length, even if 128k is inside of Llama 3.1 's pretrained context window.

2. Asymptotic memory complexity is now quadratic.

One of the most important parts of FlashAttention was reducing the memory complexity of the attention mechanism to linear from quadratic by fusing attention score computation. However, because of the requirement of top-k for each row, this method has to store an O (T^2) buffer before converting it to a sparse mask (indices). This peak memory consumption of the temporary buffer will be around 15.625GB for a 1M (Gemini's context length) sequence with 64 block sizes. And if we have to allocate additional buffers such as indices and sorted values for top-k operation (let's think the backend of top-k is merge-sort), we have to store at most 93.72GB (15.62GB for input values, 15.62GB for sorted values, 62.48 for indices). Therefore, I think authors should investigate the memory consumption of the algorithm more carefully and should investigate the way to reduce peak memory (e.g., increasing the block size of gating)

3. Limited methodological contribution.

I think the overall approach is quite similar with TDSA, but just pooling dimension is different. (hidden dim -> sequence dim). I think if there is some kind of justification or comparision between various pooling dim (hidden, head, sequence) and showing why pooling sequence dim is the most effective should be better than current form.

The reducing sequence dim approaches for generating attention masks already exist (QUEST, MInfernece), so I think we need some more justification for additional training in attention-gating networks.

4. Limited downstream task evaluation.

The only downstream task evaluation is the LongBench in Table 2. (I do not think PPL of PG19 can demonstrate actual downstream task performance because perplexity is well-known to have a weak correlation between instruction tasks.)

The context length of Table 2 is around 8 to 32k, which is within the range of their speedup claim (x5.67 on 32k). However, I think if we can show that the performance degradation is minimal or negligible in a longer context, such as 128k, the impact of this work can be more powerful (InfiniteBench: https://github.com/OpenBMB/InfiniteBench, LOFT: https://github.com/google-deepmind/loft, RULER: https://github.com/NVIDIA/RULER, etc.).

**Questions:**

- Can you show how long the training took with 400 steps? I have a huge concern about the cost, and it may be unrealistic with a long context model. I want to ask you to add FLOPs measures or wall clock GPU-hour measures of training. And the training context length is quite short compared to up-dated long context LLMs (GLM4 = 1M, Llama3.1 = 128k). Therefore, I cannot agree that training will always end in a few hours because the training time grows quadratically.

- Can you show the generalizability of the attention gating network by extending the context window further than the trained (attention gate trained context, 64k~32k)? In Table 1, attention gating tends to fail to estimate exceeding trained context length (64k), which is still inside of LLM pretrained context length.

- Can you add a hyperparameter study about block size? I think the block size of each query and key dimension will show different patterns and have different sensitivity.

---

Therefore, while I agree that the paper is well-written and organized, I think it requires more revisions. I hope we can fix the problems and improve the paper partially during the discussion period.

---

> ### Author Response · Authors · 2024-11-20
> **Response to Reviewer 4 (Require of Training Part 1)**
>
> ## Weakness 1 (Require of Training)
> SeerAttention is the first to use learning representation method that resolve the challenging **Long-context** Attention Sparsity problem in a simple but effected manner. SeerAttention can be applied in the post-training stage with minimal cost or directly used in long-context fine-tuning with more training resources.  However, we **cannot agree** that **require of training/learning is a weakness** of our work especially in this (ICLR) International Conference of **Learning Representation** conference. We understand the concern of cost but we want to emphasize that (1) a major contribution of this work is actually to make the training process very efficient (2) the capability of training makes our method a general and efficient methods, which have more potential to perform well in the long run.
>
> ### 1.We make the training of SeerAttention very efficient
>
> - **Our customized kernel make training cost as low as inference in post-training**. In Post-training usage, only the AttnGates are trained, and the original model’s weights do not require gradient. The training cost is similar to **inference** thanks to our customized kernel to generate ground truth (explained in fig3). Without this kernel, no matter using torch naive implementation or old Flash-Attention, you can not avoid the quadratic memory consumption to save the attention map to train the AttnGates.
> - **The additional RoPE in AttnGate enable its context-length extrapolation ability.** First, to clarify, a long-context LLM does not require training with data that is as long as its context window size. In most of the cases, a model is trained with both long and short context data and scaling RoPE  can also help to increase the context length. In SeerAttention, another key contribution we made is to add the additional Block RoPE in AttnGate. This RoPE helps the AttnGate to learn the block-level positional information and enable its context-length extrapolation ability similar to original LLM. **In other word, we do not necessarily need to train a 128k Model using 128k length data.** Here is a detailed experiments on Post-training cases with 500 steps different training sequence length on Llama-3.1-8B with global batchsize 16. The GPU hours is calculated as num_GPUs x num_hours.
>
> |  | MInference | MoA | Ours 8k | Ours 16k | Ours 32k | Ours 64k | 128k |
> | --- | --- | --- | --- | --- | --- | --- | --- |
> | Calibration/Post-training GPU hours | 1h | 2h (cpu) | 2h | 4h | 16h | 40h | 100h |
>
> With a single 4x or 8xA100 machine, it only takes several hours in Post-training . In addition, the Fine-tuning experiments with YaRN (32k batch size 8) in our original paper only takes 32 GPU hours.
>
> Here we also show the results of different training and evaluation context length to demonstrate that the block RoPE in AttnGate actually learns the position information. The table shows the PPL results on Llama-3.1-8B-Instruct with 70% sparsity. As shown in the table, you still can get very good performance with training on shorter data like 16k and 32k.
>
> | (with RoPE)/(without RoPE)  | training 8k | training 16k | training 32k | training 64k | training 128k |
> | --- | --- | --- | --- | --- | --- |
> | eval 8k | 10.4 / 10.8 | 10.2 / 10.7 | 10.2 / 11.3 | 10.2 / 11.5 | 10.3 / 13.6 |
> | eval 16k | 10.3 / 21.5 | 10.3 / 10.9 | 10.1 / 10.9 | 10.0 / 11.5 | 10.1 / 12.8 |
> | eval 32k | 10.4 / 31.8 | 10.3 / 17.6 | 10.3 / 11.1 | 10.2 / 11.8 | 10.2 / 15.3 |
> | eval 64k | 10.7 / 47.8 | 10.6 / 28.7 | 10.5 / 17.5 | 10.4 / 13.3 | 10.4 / 20.1 |
> | eval 128k | 12.4 / 79.4 | 11.7 / 59.3 | 11.5 / 32.9 | 11.2 / 21.4 | 11.1 / 20.5 |
> |  |  |  |  |  |  |
>
> In summary, despite our method requires training, with our customized kernel and additional RoPE design in AttnGate, it’s very efficient to train AttnGate in post-training setting with your original model untouched. **This is different from Reformer, Linear attention or Mamba-like approaches that requires you to retrain the whole network from scratch.**
>
> ### 2. Concern with LoRA
>
> To clarify, MInference and MoA do not need training, but they require offline calibration as shown in previous table. Thus, the situation of LoRA is the same. For different LoRA models, they still need to redo the calibration process. **However, as a training-based method, SeerAttention can be co-trained with any LoRA or Fine-tuning scheme just like our long-context extension experiment**. This means that your model can natively obtain the sparse capability during your LoRA or Fine-tuning, which is bound to deliver the better accuracy performance.

---

> ### Author Response · Authors · 2024-11-20
> **Response to Reviewer 4 (Require of Training part 2 + Weakness 2&3)**
>
> ## Weakness 1 (Require of Training )
> ### 3. Try Quest like reducing in Prefill
>
> Quest is mainly designed for per-Q decode. To fit in our prefill setting, we use a Quest-like design by replacing per-Q prediction with avgpooling of Q similar to SeerAttention. A major problem with Quest is that it uses element-wise max operation between Qavg*Kmax and Qavg*Kmin before reduction (summation). When hidden dimension being 128, this element-wise operation costs 128x more GPU memory compared to SeerAttention as we directly do matmul. Thus, Quest-like method is super inefficient in prefill and it gets OOM on 64k and 128k test on a single A100. We also tried adding a similar block RoPE and it improves the performance a little bit. However, it still fails to deliver reasonable accuracy. Here is the results on Llama-3.1-8B-Instruct and PG-19 with 50% sparsity.
>
> | Quest-like | 8k | 16k | 32k | 64k | 128k |
> | --- | --- | --- | --- | --- | --- |
> | PPL (with RoPE)/ (without RoPE) | 60/151 | 58/152 | 67/148 | OOM | OOM |
>
> ### 4. Generalizability
>
> First, we think the Table 1’s 128k results is not evidence that SeerAttention fails at 128k. SeerAttention still get very good results on sparsity ≤ 80%. The degradation of 90% sparsity is mainly due to the fact that we apply 90% sparsity on every single head. However, MInference allocates different sparsity ratios on different head with an averaged of 90%. We acknowledge that different head should be treated differently to get better performances. Nevertheless, a more sophisticated sparsity allocation method can be explored in the future and it is orthogonal to our current story.
>
> On the other hand, we cannot agree that training-based method does not generalize well. In our experiment, we use a single pre-trained dataset (Redpajama) and test all the downstream benchmark without seeing any generalizability issue. On the contrary, previous sparse attention methods mostly rely on human observation, which is hard to serve as a general and scalable solution.
>
> ## Weakness 2 (Memory complexity)
>
> The output of AttnGate is quadratic. However, when block size is 64, it’s only 1/4096 of the original size.  Also to clarify, our TopK is applied per-row bases instead of global TopK on the 2D map.
> Since we only need [Seq/Block, TopK] shape tensor to store indices, the memory related to TopK is much less than your calculation. (We will add more clarification in the paper.)
>
> As the computation of AttnGate is similar to self-attention (without the multiply with V part), if memory is really an issue, **we can apply exactly the same tiling strategy as FlashAttention in AttnGate**. In such case, for each local block, you can first do local TopK and then merge as global TopK in the end.
>
> ## Weakness 3 (Limited methodological contribution)
>
> Thanks for the suggestion. We have discussed many meaningful insights in ablation study. We will consider moving these earlier so that readers can better understand the contribution.
>
> In terms of TDSA, it looks like a similar approach at first glance. However, it did not solve any challenges related to long-context LLM because it reduces on hidden dimension instead of sequence dimension. For a model like Llama-3-8B, the hidden dimension is only 128 and there is not much space to squeeze. A small hidden dimension may also hurt the utilization of TensorCore in modern GPUs. However, the sequence dimension can be as large as 128K or longer. Actually, we have done an experiment using a modified version of TDSA for long context scenario, which first reduce hidden dimension and reshape the tensor in sequence dimension.
>
> ```
> q = linear(q) # [b,h,s,d] to [b, h, s, 4] reducing hidden dim to 4
> q = q.reshape(b, h, s/Block, 4 * Block) # change the reduction to seq dim similar to us
> #… linear + rope similar to us
> ```
>
> Here are the results on this modified TDSA on PG-19 with identical post-training setting as SeerAttention. It fails to deliver reasonable performance similar to Quest.
>
> | Sparsity 50% | 8k | 16k | 32k | 64k | 128k |
> | --- | --- | --- | --- | --- | --- |
> | TDSA  | 17.2 | 16.9 | 19.3 | 25.2 | 44.5 |
> | SeerAttention | 10.1 | 9.9 | 10.0 | 10.2 | 10.4 |
>
> In general, we are the first to systematically formulate and solve the Long-context sparse attention problem using a learning-based method. Similar to MoE that helps scaling the model size using learnable gates, we aim to scale the context length by using learnable AttnGate. We provide a comprehensive methodology study on:
>
> 1. How to formulate the AttnGate to address sparse long-context attention?
> 2. What’s the ground truth? How to obtain it efficiently by avoiding O(N^2) memory explosion?
> 3. How to choose Pooling methods?
> 4. How to retain the block-wise positional information after pooling?
> 5. How to train in both Post-training (only train AttnGate) and Fine-tuning settings (train AttnGate and original weights)?
>
> In summary, we have tackle many methodological or practical challenges to really make the design works.

---

> ### Author Response · Authors · 2024-11-20
> **Response to Reviewer 4 (Weakness 4 and Questions)**
>
> ## Weakness 4 (Limited evaluated task)
>
> Thanks for the suggestion of adding more downstream test. We have added experiments on ruler benchmark and compare with MInference under the same Post-training setting in the paper (Llama-3.1-8B-Instruct). In this experiment, we demonstrate that SeerAttention performs better than MInference in both accuracy and speedup. We found that MInference spends a huge amount of time doing online search of per-head sparse index on complex tasks. While our method is more  efficient and effective than MInference.
>
> |  | 4k | 8k | 16k | 32k | 64k | 128k |
> | --- | --- | --- | --- | --- | --- | --- |
> | SeerAttention | 95.5 | 92.6 | 92.7 | 87.8 | 83.3 | 71.4 |
> | MInference | 95.5 | 92.6 | 91.4 | 85.7 | 83.2 | 67.0 |
> | Speedup over MInference | 9.27x | 9.64x | 7.16x | 4.81x | 3.58x | 2.29x |
>
> ## Question 1 & 2 are answered in previous section
>
> ## Question 3 (Block Size Experiments)
>
> Here we add a simple experiment on PPL (PG-19) with block sizes 32, 64, and 128. The results come from post-training setting Llama-3.1-8B-Instruct with sparsity 70%.
>
> |  | 8k | 16k | 32k | 64k | 128k |
> | --- | --- | --- | --- | --- | --- |
> | block 32 | 10.49 | 10.36 | 10.46 | 11.09 | 11.91 |
> | block 64 | 10.18 | 10.01 | 10.10 | 10.29 | 10.71 |
> | block 128 | 10.25 | 10.04 | 10.12 | 10.31 | 10.61 |
>
> The results show that block 64 and 128 are similar and block 32 are slightly worse. We think this might be the fact when block size is larger, the sentence level information can be better preserved. However, it might be strongly related to what tasks is evaluated. We will try more experiments in our revision should you find this experiment critical.

---

> > ### Comment · Reviewer_6XFX · 2024-11-22
> > **Response to Author Rebuttal**
> >
> > First of all, thank you for a detailed and very interesting rebuttal. I think we can discuss this more with the following responses.
> >
> > ---
> >
> > > W1. 1, 2
> >
> > Thank you for your insightful and interesting rebuttal. I agree that we can do training models in ICLR. But, what I want to say is we always have to think about overall lifetime efficiency (including training time, not only inference) when we try to make something efficient methodology. That is why I want to raise the concern about training costs, which seem unscalable at this point. Since the training cost still scales linearly with the parameter scale and quadratically with sequence length, as shown in the table, I cannot agree that the current training cost is negligible or cheap.
> >
> > As an academic researcher, I think the cost of methodology is very important in the community aspect. If we require powerful GPUs, which are hard to access for many researchers outside of the industry, then how can researchers who do not have expensive GPUs conduct their research? For instance, the author said we could train the 8B model with only 40 A100 hours. But this is approximately 120 hours of A5000. This means we need to wait for 1 to 5 days to check the performance of a single test. Therefore, I think we need to discuss more about efficient training if training is necessary.
> >
> > In conclusion, I still think that the requirement of training is a significant disadvantage as an efficient attention mechanism to speed up because it effectively limits the efficiency of the model's lifecycle and further researchability. So, I want to politely raise more questions for the authors to answer: Is there any way to avoid or minimize the training cost? Can you perform an ablation study between training cost and inference performance? Are there any special patterns in training dynamics to save further training costs?
> >
> > > W1. 3
> >
> > Thank you for conducting a helpful experiment. I think the authors' claim about QUEST memory cost is wrong. As far as I understand, QUEST only requires twice as much memory to store a reduced single token from pages. However, since it groups tokens with pages, it can effectively reduce end-to-end memory costs. Did you implement it with eager mode? I think there is no reason to cost 128 times larger memory.
> >
> > > W1. 4
> >
> > Thank you for pointing out the fact that we need to handle heads differently when we try to sparsify more extremely (>90%). I think this observation is also useful for future researches.
> >
> > > W2
> >
> > This was my misunderstanding about top-k. Thank you for pointing out.
> >
> > > W3
> >
> > Thank you for pointing out many interesting stuff and adding TDSA results. These results will be helpful for further understanding how important to pool sequence dimension. I think the rebuttal resolved my concern about limited metholodolgical contribution.
> >
> > > W4
> >
> > Thank you for adding the RULER result with SeerAttention and Minference. It is impressive that SeerAttention can remain needle-finding context utilization in a long context. For this table, can you add the FA2 result? Can you add the wall-clock latency for each prefill and decoding phase for each method?
> >
> > However, I think adding NLP tasks rather than synthetic task is way much more important and valuable (that is why I ordered baselines by InfiniteBench - LOFT - RULER), therefore I want to ask author to run SeerAttention in InfiniteBench and LOFT either.
> >
> > ---
> >
> > I want to raise more questions here,
> >
> > - Q4. Can you apply the SeerAttention in the decoding phase, too? According to the conclusion section of the paper, the decoding phase uses dense rather than sparse attention. I did not notice this downside during the review. However, I think this problem is quite important to be studied.
> >
> > ---
> >
> > Again, thank you for providing a helpful rebuttal to review. I could be more confident that I understand this method now and its contribution.
> >
> > However, I think the remaining concerns are still critical and require revision. Therefore, I want to keep my rating. I hope we can resolve more concerns during the remaining discussion period.

---

> > > ### Author Response · Authors · 2024-11-22
> > >
> > > ## W123
> > >
> > > As re-emphasized and detailed explained in previous response, SeerAttention is already very cost effective in term of training AttnGate in Post-Training setting, which is as low-cost as inference. This is because you only need to calculate the ground truth of of AttnGate during forward process using our kernel with minimal memory and latency overhead (shown in Fig 8). So no matter you are an academic researcher or industrial researcher, as long as you can **inference** an Model, you can train the AttnGate (doing inference is already the lowest type of cost in LLM research). **If you believe 40 A100 hour is really a fair criteria to reject an ICLR paper, we guess over 90% of papers need to be rejected.** The advance of AI really comes from learning instead of manual feature engineering if you think of the history of computer vision and pattern recognition before CNN. It’s a whole bunch of heuristic pipeline just like the current chaos of sparse attention methods. You can observer more and more sparse pattern and design a more and more complex heuristic search method to approximate it. **This is really the type method that is not scalable.** However, as you mentioned, our method is simple but effective because it is agnostic to any prior knowledge of pattern.
> > >
> > > - **Ablation study of training cost and inference performance.** We think our previous response to your require of training part already done this. In case you miss it, we put it here again. The results show the training time cost and inference performance of different training sequence length using 500 steps. It means that you even do not need 40h A100 to train the AttnGate. **Using 16k and 32k already give you reasonable performances, which is only 12h and 48h based on your A5000 calculation.**
> > >
> > > |  | MInference | MoA | Ours 8k | Ours 16k | Ours 32k | Ours 64k | 128k |
> > > | --- | --- | --- | --- | --- | --- | --- | --- |
> > > | Calibration/Post-training GPU hours | 1h | 2h (cpu) | 2h | 4h | 16h | 40h | 100h |
> > >
> > > | (with RoPE)/(without RoPE) | training 8k | training 16k | training 32k | training 64k | training 128k |
> > > | --- | --- | --- | --- | --- | --- |
> > > | eval 8k | 10.4 / 10.8 | 10.2 / 10.7 | 10.2 / 11.3 | 10.2 / 11.5 | 10.3 / 13.6 |
> > > | eval 16k | 10.3 / 21.5 | 10.3 / 10.9 | 10.1 / 10.9 | 10.0 / 11.5 | 10.1 / 12.8 |
> > > | eval 32k | 10.4 / 31.8 | 10.3 / 17.6 | 10.3 / 11.1 | 10.2 / 11.8 | 10.2 / 15.3 |
> > > | eval 64k | 10.7 / 47.8 | 10.6 / 28.7 | 10.5 / 17.5 | 10.4 / 13.3 | 10.4 / 20.1 |
> > > | eval 128k | 12.4 / 79.4 | 11.7 / 59.3 | 11.5 / 32.9 | 11.2 / 21.4 | 11.1 / 20.5 |
> > >
> > > ## W1.3
> > >
> > > Quest only works for decode. If you take a deeper look on in our Quest experiment, what we do is extending original Quest method in Prefill under similar setting of SeerAttention, i.e., predicting block-level sparsity of self-attention in Prefill. As a results, if you first do a element wise max comparison between hidden dim, you need to compute Qavg* Kmax ([b, h, s/block, s/block, dim]) and Qavg* Kmin (also shape [b, h, s/block, s/block, dim]), performing max (also resulting [b, h, s/block, s/block, dim]) and then reduction ([b, h, s/block, s/block]). You need to store this huge tensors with hidden dim to do element-wise max. However, our method performs matmul(Qpool, Kpool.T), which directly outputs a [b,h, s/block, s/block] tensor. That’s why Quest-like reduction does not work efficiently in Prefill.
> > >
> > > ## W4
> > >
> > > - Most of the time Ruler only outputs only a few generation tokens, so the major part is in prefill.
> > > - We feel quite confused of this request in adding infinitebench and LoFT. What you said last time is “ **However, I think if we can show that the performance degradation is minimal or negligible in a longer context, such as 128k, the impact of this work can be more powerful (InfiniteBench，LOFT, RULER, etc.).“** We think this generally means the purpose of this experiment is used to demonstrate that SeerAttention works well in 128k challenging task. And our ruler test already prove this. If you do care about InfiniteBench, why not specifically asking for InfiniteBench at the beginning. We are sorry that we can not accept this request as there are all kinds of different tasks like this and we feel this is an endless discussion on which one is more important. **In fact, we already include LongBench in our experiment, which is a real-world  task instead of synthetic one.**
> > >
> > > ## Decode
> > >
> > > We have not done experiment on decode yet and this is left for future work as we mentioned in discussion. We can not agree that this is a downside. Previous works like MInference only works for Prefill, while Quest, H2O only work for decode. They are indeed two different system problem (One is compute bound and one is memory bound). In fact, because of the quadratic scaling, long-context perfill is usually the bottleneck in latency. Decode time is only linear scaled wrt the context length. We acknowledge that we should re-emphases more in the paper/title but we can not agree this is a downside.

---

> > > ### Author Response · Authors · 2024-11-22
> > > **Summary of our second response**
> > >
> > > ## Summary
> > >
> > > ICLR is only 10 pages. We think fundamentally what we want to say is that we are the first to bring this revolution of sparse attention using simple learning-based method. Thanks to your previous suggestions, we have already made great improvement in  1) training cost, 2) methodology contributions 3) additional downstream tasks. We think adding more downstream tasks and decode experiment do not fundamentally make too much difference. Besides, even through you are not interested in fine-tuning, those who are deeply invested in LLM serving might be more interested in this part. Restricting ourselves solely to the post-training phase actually limits the potential benefits of this work.

---

### Official Review · Reviewer_niUN · 2024-11-03

**Soundness:** 3
**Presentation:** 3
**Contribution:** 2
**Rating:** 5
**Confidence:** 4

**Summary:**

This paper introduces a learnable method to leverage attention sparsity, enhancing the efficiency of large language models (LLMs) in long-context inference. Specifically, the authors developed a learnable gate that adaptively selects significant blocks within the attention map, treating non-significant blocks as sparse. Furthermore, to facilitate efficient training of this gate, the paper incorporates a customized FlashAttention implementation that efficiently extracts block-level ground truth from the attention map with minimal overhead. Experimental results demonstrate that the proposed technique achieves a superior accuracy-efficiency trade-off when processing long-context inputs compared to baseline approaches.

**Strengths:**

- Addressing the accuracy-efficiency trade-off of LLMs during long-context inference is a critical challenge, especially given the recent trend of employing LLMs to tackle increasingly complex problems with sophisticated inference processes.

- The proposed approach, which learns a sparse attention distribution rather than relying on pre-defined attention patterns or heuristics to approximate sparsity, is intuitively effective in achieving a superior accuracy-efficiency trade-off.

- The customized FlashAttention modification enhances real-device efficiency, providing tangible performance improvements over the naïve implementation and reinforcing the feasibility of applying this approach in real-world applications.

- The paper is well-organized, making it accessible and straightforward to read and comprehend.

**Weaknesses:**

- **Limitations in Related Work Discussion**: Alleviating attention sparsity to enable long-context inference through attention optimization has been a significant area of research, with numerous discussions surrounding it. To provide a more comprehensive background, it would be beneficial for the authors to include additional related works on attention sparsity, particularly approaches that leverage KV cache eviction, encompassing both pre-defined patterns [1] and dynamic pattern attention sparsity [2,3,4].

- **Limited Context Length Evaluated**: Recent works utilizing attention optimization to enhance long-context inference capabilities have successfully extended the context length to settings up to 1M, significantly exceeding the pretrained context limits of LLMs, even without training adjustments [1,2]. However, this paper evaluates only up to a 128K context length. Given that the pretraining context length of the target Llama-3.1 model is also 128K, further experiments with longer context windows would help elucidate the proposed approach’s potential in handling extended contexts.

- **Deployment Cost of the Proposed Approach**: While existing approaches aim to improve the long-context capabilities of LLMs without requiring additional training by exploiting attention map sparsity, this work entails additional model tuning. Benchmarking both the post-training and fine-tuning variants of the proposed approach against existing training-free techniques would offer greater insight into the effectiveness of the required additional training.

- **Setting Mismatch Between Training and Memory Profiling**: In the pooling selection section, the authors state that they use average pooling for Q and a combination of max and min pooling for K. However, in the memory profiling results presented in Figure 8, only max pooling is considered in the training kernel. Would incorporating average pooling, as implemented in the proposed approach, significantly increase memory overhead? Additionally, what would the memory overhead look like during the inference process? How does the additional training introduced by this approach impact memory usage?

[1] Xiao, Guangxuan, et al. "Efficient streaming language models with attention sinks." arXiv preprint arXiv:2309.17453 (2023).

[2] Zhang, Zhenyu, et al. "H2o: Heavy-hitter oracle for efficient generative inference of large language models." Advances in Neural Information Processing Systems 36 (2024).

[3] Ge, Suyu, et al. "Model tells you what to discard: Adaptive kv cache compression for llms." arXiv preprint arXiv:2310.01801 (2023).

[4] Wang, Zheng, et al. "Model tells you where to merge: Adaptive kv cache merging for llms on long-context tasks." arXiv preprint arXiv:2407.08454 (2024).

**Questions:**

- **Theoretical or Intuitive Explanation for Pooling Selection**: In the pooling selection section, the proposed approach utilizes average pooling on Q and a combination of max and min pooling on K. Could the authors provide a more theoretical or intuitive explanation for this choice? Specifically, given that many existing studies have highlighted the presence of excessively high attention scores on tokens with limited semantic information [1,2,3], could this phenomenon support the rationale behind the pooling selection?

- **Row-Normalization of Max-Pooled Attention Map**: In Section 4.1, the authors mention scaling the max-pooled attention map by row-normalizing it to sum to 1. Could the authors further elaborate on this step? As previous studies [4] suggest, it is critical to maintain the values of the attention distribution and avoid out-of-distribution attention scores (i.e., scores that are too large or too small) for informative tokens. Given this, might the proposed normalization inadvertently produce out-of-distribution attention scores? Have the authors experimented with alternative approaches, such as leaving the attention map unaltered or redirecting unused attention to the first attention sink token?

- **Applicability to Other LLM Architectures**: Can the proposed approach be adapted for other types of large language models, such as models based on the Mixture of Experts (MoE) architecture?

[1] Yu, Zhongzhi, et al. "Unveiling and harnessing hidden attention sinks: Enhancing large language models without training through attention calibration." arXiv preprint arXiv:2406.15765 (2024).

[2] Sun, Mingjie, et al. "Massive activations in large language models." arXiv preprint arXiv:2402.17762 (2024).

[3] Ge, Suyu, et al. "Model tells you what to discard: Adaptive kv cache compression for llms." arXiv preprint arXiv:2310.01801 (2023).

[4] Xiao, Guangxuan, et al. "Efficient streaming language models with attention sinks." arXiv preprint arXiv:2309.17453 (2023).

---

> ### Author Response · Authors · 2024-11-20
> **Response to Reviewer 3 (Weakness 1-3)**
>
> ## Weakness 1 (Limitations in Related Work Discussion):
>
> Thanks for pointing out these related works. We will add more discussion in our revision.
>
> ## Weakness 2 (Limited Context Length Evaluated):
>
> Thanks for the suggestion of increasing the evaluation length. Since you mentioned StreamingLLM can extend the context length of a model without training, we perform an additional needle-in-a-haystack experiments in ruler benchmark and found that StreamLLM can not perform well in this challenging task. Here is the average score on Llama-3.1-8B-Instruct under post-training setting.
>
> | Average Score of 4k-128k | single  | multi |
> | --- | --- | --- |
> | SeerAttention | 100 | 94.7 |
> | MInference | 99.3 | 94.5 |
> | Streaming LLM | 82.4 | 56.7 |
>
> This is mainly because that not all attention heads demonstrate this attention sink pattern in StreamingLLM, which highlight the advantage of our method that is agnostic to any pre-defined sparse pattern. We did not include [1,2] in our results comparison because they are relatively earlier works and our baseline MInference and MoA already surpass them.
>
> In terms of 1M context length, it requires additional system optimization like memory offloading, tiling, and etc to avoid OOM on single A100, which is orthogonal to our current study. However, we did a simple test with up to 0.5 million context length using gradientai/Llama-3-8B-Instruct-Gradient-1024k model. Here is the PPL results on PG19 (filtered with length ≥ 0.5M).
>
> |  | 128k | 256k | 512k | 1M |
> | --- | --- | --- | --- | --- |
> | baseline | 9.2 | 8.9 | 8.5 | OOM |
> | sparsity 50% | 9.5 | 9.2 | 9.6 | OOM |
>
> Due to the time limitation of rebuttal, we can not fully explored the optimization of training and system implementation of 1M model. However, we think there are great opportunities for SeerAttention to apply in longer context models.
>
> ## Weakness 3 (Deployment Cost of the Proposed Approach):
>
> To clarify, methods like MInference and MoA do not need training, but they also require offline calibration. In SeerAttention, we have made two contributions to make our deployment cost low. On the one hand, we customize a training kernel to generate ground truth, which avoid the quadratic memory issue of storing attention map (shown in Fig3). On the other hand, we add a block-level RoPE in AttnGate to enable context-length extrapolation ability of AttnGate.
>
> Here is the GPU hours (num_GPU * num_hours) statistics. For SeerAttention, we use a batch size of 16 with 500 training steps.
>
> |  | MInference | MoA | Ours 8k | Ours 16k | Ours 32k | Ours 64k | 128k |
> | --- | --- | --- | --- | --- | --- | --- | --- |
> | Calibration/Post-training GPU hours | 1h | 2h (cpu) | 2h | 4h | 16h | 40h | 100h |
>
> As for fine-tuning experiment with YaRN in the paper, it takes 32 GPU hours to train with 32k data with batch size of 8 for 400 steps.
>
> We show that our block-level RoPE design in AttnGate enables the ability of to learn the block postition information. This means that you do not have to use data as long as your context window to train your AttnGate. Here is the PPL results with different training context length with and without block RoPE in AttnGate (a more detailed version of Fig9).
>
> | (with RoPE)/(without RoPE) | training 8k | training 16k | training 32k | training 64k | training 128k |
> | --- | --- | --- | --- | --- | --- |
> | eval 8k | 10.4 / 10.8 | 10.2 / 10.7 | 10.2 / 11.3 | 10.2 / 11.5 | 10.3 / 13.6 |
> | eval 16k | 10.3 / 21.5 | 10.3 / 10.9 | 10.1 / 10.9 | 10.0 / 11.5 | 10.1 / 12.8 |
> | eval 32k | 10.4 / 31.8 | 10.3 / 17.6 | 10.3 / 11.1 | 10.2 / 11.8 | 10.2 / 15.3 |
> | eval 64k | 10.7 / 47.8 | 10.6 / 28.7 | 10.5 / 17.5 | 10.4 / 13.3 | 10.4 / 20.1 |
> | eval 128k | 12.4 / 79.4 | 11.7 / 59.3 | 11.5 / 32.9 | 11.2 / 21.4 | 11.1 / 20.5 |
>
> In summary, you can achieve reasonable performance by only spending a small amount of training resources.

---

> ### Author Response · Authors · 2024-11-20
> **Response to Reviewer 3 (Weakness 4 and Questions 1-3)**
>
> ## Weakness 4 (Fig8 clarification):
>
> Fig8 reflects the performance kernel that we use to generate the 2D maxpool attention map kernel for training AttnGates as illustrated in Fig2(b). For clarification, this is not the kernel to run AttnGate, but the kernel to generate the ground truth to train AttnGate. We want to use Fig8 to show that with the help of this kernel, our training process can be very efficient compared to manual torch implementation. This is because we modify flash-attention kernel and do not need to explicated generate the O(N^2) full attention map.
>
> **Memory overhead in inference:**
>
> The major overhead of our methods lies in the AttnGate outputs because it’s the only quadratic part. For a layer with 32 heads like llama-8B, the extra tensor shape is  [32, seq/B, seq/B]. For a model with 128K seq length. The extra cost is only about 0.25GB. For very extremely case like 1M model, you can tile the AttnGate computation similar to FlashAttention to save memory if necessary.
>
> **Memory overhead in training:**
>
> The memory cost of post-training method is quite small, since we do not need to compute the gradient of original weights and LM loss (logits). You can easily train with 128k sequence data without any fancy trick to save memory like tensor parallelism or pipeline parallelism on a single A100.
>
> In terms of fine-tuning, we also customized a block sparse flash-attn kernel for backward (not discussed in the paper). As a result, the major memory consumption part is similar to dense fine-tuning baseline. The overhead of AttnGate is similar to inference case mentioned above.
>
> ## Question 1 (Theoretical or Intuitive Explanation for Pooling Selection):
>
> We have provided a short discussion sec 6 (Pooling Ablation). We think that K prefers max min while Q prefers avg can be related to a well-known phenomenon in Quantization field that K has more outliers than Q and V.  As a results, we think the outliers in K might carry more information and thus are better preserved by max-min pooling.
>
> ## Question 2: (Row-Normalization of Max-Pooled Attention Map):
>
> The Row-Normalization of max-pooled Attention map is mainly used to adhere to the output distribution of AttnGate in training where softmax is used (row sum = 1). Currently we found this ground truth works well and have not tried other methods. We tested by using this generated ground truth to run different benchmarks under high sparsity ratios and it gives nearly no accuracy degradation. This means that as long as the trained AttnGate can faithfully learn from the ground truth, it will perform well.
>
> On the other hand, you can still combine AttnGate with other method like StreamingLLM that always requires it to attend to first token. However, we did try this before but did not see much accuracy gain. In fact, those head with distinct streaming pattern can be learned easily by our AttnGate as shown in Fig7.
>
> ## Question 3: (Compatible with MoE)
>
> Since MoE works in MLP layers and SeerAttention works in Attention layers, we can definitely combine with each other. Here are the PPL results on Mixtral-8x7b MoE model, which validate that SeerAttention can be effectively applied to MoE.
> | PPL | 8k | 16k | 32k |
> | --- | --- | --- | --- |
> | dense | 6.04 | 6.05 | 6.00 |
> | sparsity 0.5 | 6.07 | 6.08 | 6.02 |
> | sparsity 0.6 | 6.10 | 6.10 | 6.04 |
> | sparsity 0.7 | 6.14 | 6.13 | 6.07 |
> | sparsity 0.8 | 6.22 | 6.18 | 6.11 |
> | sparsity 0.9 | 6.47 | 6.30 | 6.21 |

---

### Official Review · Reviewer_5GA1 · 2024-11-04

**Soundness:** 2
**Presentation:** 2
**Contribution:** 3
**Rating:** 5
**Confidence:** 4

**Summary:**

This paper presents SeerAttention, a attention mechanism learning intrinsic sparse attention in LLMs. It outperforms existing methods in post-training and fine-tuning, with efficient implementation and good adaptability to different context lengths and sparsity ratios.

**Strengths:**

- The proposed SeerAttention mechanism learns attention sparsity instead of relying on predefined patterns. This allows for better adaptation to different language tasks and models, as demonstrated by its performance across various experiments.
- The development of a customized FlashAttention implementation enables efficient learning of the gating network by extracting the block-level ground truth of the attention map with minimum overhead. This not only improves the training process but also contributes to the efficiency of the model.

**Weaknesses:**

- In section 3.1, the description is insufficient. It is hard to understand the proposed method only with the description in Section 3.1, but I found Figure 2 is easy to understand. I suggrest that more details around Figure 2 should be added in Section 3.1 to enhance understanding. For instance, the operations and significance of each component in the AttnGate module need to be elaborated.
- In section 4.2, many symbols related to FlashAttention are used without prior explanation, making it difficult for readers unfamiliar with FlashAttention to follow.
- The meaning of "max and min pooling on K matrix" is not clear. It should be precisely defined whether it is applied to different dimensions or in a specific way.
- When seq cannot be evenly divided by B (as the size of Q and K after downsampling is [seq/B,d]) during the growth of seq in token generation, the proposed method does not adequately discuss how to handle these situations. It is also unclear if the method is only applicable to the Prefilling stage and not suitable for the token decoding stage.
- After reading the paper, I found that only the blocks with larger attention scores from Atten Gate are used for calculation with flash attention during infernece. If an inference stage Triton kernel is implemented, its pseudo code should be included in the paper.
- There are inconsistencies in algorithm symbols, such as the use of A in Figure 3 and D in the appendix to represent Attention Score, which is confusing.
- Using only a part of the blocks for attention calculation raises questions about information retention. In very long sequences, discarding a large number of tokens may affect the model's ability to perform various text tasks. Since most experimental results report PPL metric, which may not be sensitive to token information sufficiency, additional experiments like Needle-In-A-Haystack or PassKey Retrivel could be added for a more comprehensive evaluation.
- It is not clear if the results in Figure 8 report to Prefilling or decoding stage delays.
- The data requirements of SeerAttention during Post-training and Fine-tuning are not specified.
- The inference delay of the model could be reported in Table 1 for a more complete comparison.

**Questions:**

NA

---

> ### Author Response · Authors · 2024-11-20
> **Response to Reviewer 2**
>
> ## Weakness 1 (Improve figures of Section 3.1):
>
> Thanks for the suggestion, we have added a dataflow figure similar to Figure 2 that explains the AttnGate operations (see Figure 11 in appendix in our rebuttal revision pdf).
>
> ## Weakness 2 (FlashAttention symbol unexplained):
>
> Thanks for the suggestion, we will add more explanation of related symbol in sec 4.2. We will keep faithfully the symbols used in FlashAttention such that readers can always refer to the original paper.
>
> ## Weakness 3 (The meaning of "max and min pooling on K matrix" is not clear):
>
> Thanks for pointing out this issue. We will improve the wording and figure caption. The pooling operation happens in the Seq dimension. For example, original Q tensor shape is [Batch, Head, Seq, Dim], the pooled result will be  [Batch, Head, Seq / Block, Dim]. When more than one pooling method are used like Max+Min pooling in K, the tensor will first be concated in hidden dim before the linear layer ( [Batch, Head, Seq / Block, Dim * 2] in this case).
>
> ## Weakness 4 (Rounding Issue when Seq not evenly divided by B):
>
> When seq not evenly divided by B, the standard way is to use padding. In fact, the pooling function in torch can automatically support “padding” by using ceil_mode = True.
>
> In current work, we focus on accelerating Prefill stage and leave decode for future exploration as mentioned in discussion section. Since Prefill and Decode are distinct two phases, many previous works also only focus on one side. For example, MInference merely focuses on Prefill, while Quest, H2O only focus on Decode.
>
> Our work has potential to also be applied to Decode phase (doing per-Q token AttnGate without pooling). In general, the distribution of average pooling on Q tensor is similar to each token’s Q vector. As a result, we can enable similar training strategy to train AttnGate with generated tokens on the same linear layer’s weights. This is left for future work.
>
> ## Weakness 5 (Pseudo code of Inference Kernel):
>
> Thanks for the suggestion. We will add pseudo code of our inference kernel as well. A very simple version will be like this:
>
> ```python
> Input: Q, K, V; Output: O, A
> for i from 1 to Tr
> 	Qi = load(Q, i) # Load Q block to HBM
> 	acc = zeros() # Output buffer
> 	for j from 1 to TopK # Only Topk block to compute
> 		Kj_index, Vj_index = k_index[j], j_index[j] # Spare BLock Index
> 		Kj, Vj = load(K, Kj_index), load(V, Vj_index) # Spare load KV
> 		sij = dot(Qi,Kj), rij = rowmax(sij)
> 		mi = max(mi, rij)
> 		p = exp(sij - mi)
> 		lij = sum(p, dim=1)
> 		alpha = exp(mi - rij)
> 		li = li * alpha + lij
> 		acc = acc * alpha + dot(p, Vj)
> 	acc /= li
> 	store(acc, O, i) # Store Acc to Output
> 	Return O
>
> ```
>
> ## Weakness 6 (Inconsistencies in algorithm Symbol):
>
> Thanks for pointing out this detail mistake, we will make it consistent.
>
> ## Weakness 7 (Evaluation in other task):
>
> To clarify, we do not discard tokens. We skip unimportant blocks in Attention computation, which is different than completely drop this token in the input. We also add new experiment on ruler, which contains various types of needle-in-a-haystack test. Here is the average score under llama-3.1-8B-instruct under post-training setting:
>
> |  | SeerAttention | MInference | StreamingLLM |
> | --- | --- | --- | --- |
> | Average | 96.7 | 96.3 | 66.4 |
>
> The results shows that our method is better than MInference and StreamingLLM (mentioned by other reviewers).
>
> ## Weakness 8 (Figure 8 meaning):
>
> Figure 8 shows the performances of our kernel for **training** (the kernel that generate the max-pooled attention map as ground truth).  This kernel is not used in inference, so no concept “prefill” or “decode”. We use figure 8 to show that our training process is very efficient so that our model can be trained end-to-end by using this kernel. This makes the overall post-training cost as low as inference.
>
> Besides, the inference kernel results lie in fig5 fig6 and table4. We will highlight **“training/inference”** our figure title and caption for clarity.
>
> ## Weakness 9 (What training data is used):
>
> We have mentioned in Sec5 line 280 that we use Redpajama dataset, which is the training dataset we use in all experiments (post-training and fine-tuning). Redpajama dataset is widely used in other long-context training works like LongLora and Longrope. We did not use any training data related to specific benchmark or test.  For future work, we believe by doing SFT-like training, SeerAttention can achieve better performance in instruct-following tasks and also enable the sparse decoding ability.
>
> ## Weakness 10 (Merge Table 1 and Table 4):
>
> Thanks for the suggestion. We will consider merging the two tables. However, the resulting table might be huge and unclear to reader. Separating the table into accuracy and efficiency evaluation might better fit our current flow.

---

> > ### Comment · Reviewer_5GA1 · 2024-11-24
> > **More Questions**
> >
> > In pre-filling phase of inference, the length of the input sequence is dynamic due to the uncertain user prompt. In this case, the authors state they can add some paddings (often using left padding) to make the sequence length be divided by B. My question is that in the first block, the pooling would be conducted over both padding tokens and user prompt tokens, what the meaning of the pooling results? Or could the proposed method handle this case?
> >
> > In Figure 6, I find only when sparsity is greater than 0.6, the speedup become significant. But in Table 2, the results on LongBench only include the variants with sparsity<=0.5. I know in Table 1, the authors report the performance with high sparsity, but PPL is not direct metric and hard to reflect the true performance of the model.

---

> > > ### Author Response · Authors · 2024-11-24
> > > **Reply to More Questions**
> > >
> > > ## More about padding
> > > We do not have to pad the **input sequence** like you mentiond. Instead, **torch pooling function supports ceil-mode** which directly "pad" in the pooling function.
> > > For example, suppose you have a input length of 7 tokens, and the block size is 4 with avgpooling. Instead of padding the input sequence to 8, we directly process the 7 tokens input. With the ceil-model in avgpooling function, the output size is 2 = ceil(7/4). So the first 4 tokens will be averged as the first output and the remanining 3 tokens will be averaged as the second output. Similar operation can be found in max and min pooling. For more details in ceil-mode, you can refer to torch mannul (https://pytorch.org/docs/stable/generated/torch.nn.AvgPool2d.html). As a result, we do not have to pad the input sequence.
> > >
> > > ## Longbench Sparsity
> > > We only chose a max of 0.5 sparsity since our baseline MoA and MInference do not have results in >50% sparsity. We have just added experiments with larger sparsity and please wait for tomorrow's results.
> > >
> > > ## Speedup ratio
> > > The current block-sparse attention kernel is implemented in Triton, which is known to have around 70% performance compared to CUDA implemenation. If compared to Triton dense flash-attention implemenation, our sparse version can achieve very linear speedup wrt sparsity. But we use CUDA's version of flash-attention in our paper as baseline.
> > > This Triton kernel mainly works as a proof-of-concept implemenation and we will implement our own CUDA kernel in the future or integrate with some 3rd-party open-sourced sparse kernels (e.g. https://github.com/mit-han-lab/Block-Sparse-Attention).

---

> > > ### Author Response · Authors · 2024-11-25
> > > **LongBench Results**
> > >
> > > Here are the results of Longbench with sparsity levels ranging from 0.6 to 0.8. It is observed that for longer inputs, the decrease in accuracy is less pronounced with higher sparsity. The overall trend is similar to the ppl results in fig4.
> > >
> > > | LongBench | 0-4k | 4-8k | 8k+ |
> > > | --- | --- | --- | --- |
> > > | sparsity 0.6 | 52.75 | 52.68 | 52.23 |
> > > | sparsity 0.7 | 51.26 | 51.65 | 51.76 |
> > > | sparsity 0.8 | 47.53 | 50.43 | 50.73 |
> > >
> > > Hope the response address your concerns.

---

> > > > ### Comment · Reviewer_5GA1 · 2024-11-28
> > > > **More comments**
> > > >
> > > > I think the performance drop with sparsity>= 0.8 is very large, so I decide to maintain my scoring. Or could the authors can demonstrate that the proposed method with sparsity= 0.8 or 0.9 outperforms SOTA methods in terms of both longbench average score and  inference speed.

---

> ### Author Response · Authors · 2024-11-28
> **Longbench results**
>
> If you combine the orginal results, you can find that we already outperforms SOTA methods (MoA and MInference).
> - MoA: MoA needs to predefine sparsity ratio. Our sparsity=0.5 results are all higher than MoA's sparsity=0.35 results.
> - MInference: It does not support user-defined sparsity. It directly searches a sparsity for you. As we explain before, short context data have less sparsity. In order to preserve accuracy, MInference only use very small sparity. For 0-4k, it uses an avg of 0.06 sparsity and our sparsity=0.1 is higher than MInferece (55.91 > 55.23). For 4-8k, its avg sparsity is 0.25, our sparity=0.25 is better than MInference (54.09 > 53.87). For 8k+, it uses an avg of 0.45 sparsity, our score is also higher when sparsity=0.5 (52.43 > 52.18).
>
> In summary, under higher or similar sparsity ratio. We achieve better accuracy results.
> In terms of speedup, please refere to Fig6, where we show that our kernel speedup is better than those works under different sparsity and context length.
>
> |  | 0-4k | 4-8k | 8k+ |
> | --- | --- | --- | --- |
> | MoA (sparsity 0.35) | 50.74 | 49.84 | 51.89 |
> | MInference | 55.23(sparsity=0.06) | 53.87(sparsity=0.25) | 52.18(sparsity=0.45) |
> | sparsity 0.1 | 55.91 | 54.32 | 53.28 |
> | sparsity 0.25 | 55.00 | 54.09 | 52.22 |
> | sparsity 0.5 | 52.40 | 52.85 | 52.43 |
> | sparsity 0.6 | 52.75 | 52.68 | 52.23 |
> | sparsity 0.7 | 51.26 | 51.65 | 51.76 |
> | sparsity 0.8 | 47.53 | 50.43 | 50.73 |

---

### Official Review · Reviewer_dwG2 · 2024-11-06

**Soundness:** 3
**Presentation:** 3
**Contribution:** 3
**Rating:** 6
**Confidence:** 4

**Summary:**

This work introduces SeerAttention, which uses pooled embedding to compute attention block mask and uses the mask to do block sparse attention for reduced latency. To enable efficient learning of the gating network, a customized FlashAttention kernel is implemented that extracts the block-level ground
truth of attention map with minimum overhead. Experiments show that SeerAttention can achieve 90% sparsity ratio at 32k context length and offere 5.67x speedup over FlashAttention-2.

**Strengths:**

* SeerAttention operates at the block level, leading to high efficiency potential.
* SeerAttention learns the sparsity pattern during fine-tuning, which is more flexible than heuristic sparsity.
* An efficient SeerAttention kernel is provided and leads to a 5+x speedup over FlashAttention.

**Weaknesses:**

* I understand that SeerAttention allows users to adjust the balance between sparsity and accuracy. However, how to determine the sparsity (or Top-k) in practice is not clear to me. I suggest the authors include a discussion on how to choose sparsity in order to maintain high accuracy.

**Questions:**

* What if the pooling is done on the RoPE'd QK embeddings? Do we still need the RoPE in the AttnGate?
* I assume that using MSE loss to train AttnGate requires fewer training steps and samples. However, what if we train AttnGate using end-to-end LM loss?
* What is the sparsity pattern? Does the max-pooled attention pool prefer recent tokens (like sliding window attention) or does it selectively preserve "important tokens" (like needles in the haystack)?
* Will SeerAttention impact general benchmarks like MMLU?

---

> ### Author Response · Authors · 2024-11-20
> **Response to Reviewer 1**
>
> ## Weakness 1 (How to choose sparsity?):
>
> Thanks for the suggestion. Currently, we expose the value of K (TopK) to users, allowing them to adjust it freely based on their targeted accuracy-efficiency tradeoff. We can incorporate offline calibration methods that search for a “recommended” sparsity ratio allocation strategy based on **input sequence length** or **complexity of input question**. For example, in the newly added MMLU test below, we use a simple linear strategy based on **input sequence length**. However, more sophisticated strategies with per-layer/head adjustments are left for future exploration.
>
> It should be noted that previous works like MoA and MInference cannot adjust the sparsity ratio as flexibly as SeerAttention. For example, MoA and MInference need to search for a pre-defined sparsity pattern offline, which do not support changing the sparsity ratio by user during runtime. In SeerAttention, since the output of AttnGate provides a “soft” score for each block, it has the **ability** to apply different TopK ratios.
>
> ## Question 1 (Why RoPE in AttnGate?):
>
> Please refer to section 6 line 482 and figure 9, where we did a RoPE ablation study that address your question. If we directly use RoPE’d QK, the AttnGate loss the ability of **context length extrapolation**.
>
> Here is a more detailed version of the study on Llama-3.1-8B-Instruct:
>
> | PPL (with RoPE)/(without RoPE) | training 8k | training 16k | training 32k | training 64k | training 128k |
> | --- | --- | --- | --- | --- | --- |
> | eval 8k | 10.4 / 10.8 | 10.2 / 10.7 | 10.2 / 11.3 | 10.2 / 11.5 | 10.3 / 13.6 |
> | eval 16k | 10.3 / 21.5 | 10.3 / 10.9 | 10.1 / 10.9 | 10.0 / 11.5 | 10.1 / 12.8 |
> | eval 32k | 10.4 / 31.8 | 10.3 / 17.6 | 10.3 / 11.1 | 10.2 / 11.8 | 10.2 / 15.3 |
> | eval 64k | 10.7 / 47.8 | 10.6 / 28.7 | 10.5 / 17.5 | 10.4 / 13.3 | 10.4 / 20.1 |
> | eval 128k | 12.4 / 79.4 | 11.7 / 59.3 | 11.5 / 32.9 | 11.2 / 21.4 | 11.1 / 20.5 |
>
> Results show that our AttnGate can learn block-wise postition information effectively with the block-wise RoPE.
>
> ## Question 2 (Currently use MSE loss, how about e2e LM loss?):
>
> Yes, the MSE loss can help the AttnGate converge fast and we also believe using end-to-end LM loss have better potential than MSE. However, this requires additional effort such as making operation like TopK and attention mask differentiable, which is left for future exploration.
>
> ## Question 3 (What’s the sparsity pattern):
>
> Please refer to Sec6 Figure 7, where we visualized some AttnGate’s outputs. In fact, without any human prior knowledge, it can automatically infer common sparsity pattern you mentioned: sliding window, attn sink, vertical & slash, and etc. This highlight SeerAttention’s ability as  a more general and scalable method than manual sparse feature engineering as the sparsity pattern may change over different models architecture, training data, context and etc.
>
> ## Question 4 (MMLU experiment):
>
> We have done additional experiment on MMLU, which is a short context-length benchmark (input sequences range from 200 to 3000 tokens). In short input like this, attention layer is not the bottleneck in inference and enforcing large sparsity may hurt the accuracy. As a results, we use a  very simple linear sparsity allocation method: input with seqlen 3000 uses 80% sparsity and no sparsity when input seqlen is 0 (the equation will be sparsity = 0.8/3000 * seqlen). Under this setting, the MMLU (5 shots) test results got similar performance as dense baseline.
>
> | Dense Baseline | SeerAttention |
> | --- | --- |
> | 0.6815 | 0.6820 |

---

> > ### Comment · Reviewer_dwG2 · 2024-11-25
> >
> > Thanks for the responses. The 'linear strategy based on input sequence length' and 'MMLU results' really strengthen the paper. Please include them in the final version. Additionally, I have a few follow-up questions regarding Q1/3.
> >
> >
> >
> > **Q1**
> >
> > > If we directly use RoPE'd QK, the AttnGate loses the ability for context length extrapolation.
> >
> > Does this mean that pooling will diminish the RoPE embedding and therefore prevent the model from extrapolating?
> >
> > **Q3**
> >
> > Can the authors provide more plots/analysis showing the attention mask patterns across attention heads and layers? For example, how do shallow and deep layers differ in their attention patterns? Within a single layer, what is the proportion of different types of heads? This additional analysis would help the community better understand attention mechanisms and amplify the impact of this work.

---

> ### Author Response · Authors · 2024-11-26
> **2nd Response**
>
> ## Q1 More about in AttnGate's RoPE
> Here is a more detailed analysis similar to fig9 in the paper. We train and evaluate different context length. It turns out that without the additional RoPE in AttnGate (Directly use the RoPE'd Q K), the evaluation length are limited (overfitting to training length). We think this is because the RoPE properties are damaged after pooling in sequence dimention. And using the non-RoPE'd QK for pooling with the additional block-RoPE in AttnGate can solve this problem.
>
> | PPL (with RoPE)/(without RoPE) | training 8k | training 16k | training 32k | training 64k | training 128k |
> | --- | --- | --- | --- | --- | --- |
> | eval 8k | 10.4 / 10.8 | 10.2 / 10.7 | 10.2 / 11.3 | 10.2 / 11.5 | 10.3 / 13.6 |
> | eval 16k | 10.3 / 21.5 | 10.3 / 10.9 | 10.1 / 10.9 | 10.0 / 11.5 | 10.1 / 12.8 |
> | eval 32k | 10.4 / 31.8 | 10.3 / 17.6 | 10.3 / 11.1 | 10.2 / 11.8 | 10.2 / 15.3 |
> | eval 64k | 10.7 / 47.8 | 10.6 / 28.7 | 10.5 / 17.5 | 10.4 / 13.3 | 10.4 / 20.1 |
> | eval 128k | 12.4 / 79.4 | 11.7 / 59.3 | 11.5 / 32.9 | 11.2 / 21.4 | 11.1 / 20.5 |
>
>
> ## Q2 Visualization of Sparse Pattern
> Thanks for the suggestion, we will add more visualization in the appendix should it get accepted. However, a 8B model has 32x32 =1024 heads, we can not draw all of them. We will open-source the visualization code to help the community to better study the attention mechanism.
>
> Here are some empirical findings that we have observed:
>  - A pattern of a head might change wrt different input. This is the reason that we can not simply categorize whether a head is pure streaming or vertical head at this stage. It also highlights our method that might be more adaptive to different pattern on a single head.
>  - For each layer, there is a portion of heads that demonstrates more random/dynamic patterns than the others.
>
> In summary, we can add a small section in appendix to discuss some empirical findings. And we will open-source the visualiation code to allow people studying the attention patterns.

---

> ### Comment · Reviewer_dwG2 · 2024-12-02
>
> Thank you for the reply. It answers some of my questions. I have raised my score.

---

### Meta-Review · Area_Chair_iuKq · 2024-12-21

**Metareview:**

**Summary:** The paper introduces SeerAttention, which exploits block-level sparsity in attention maps by learning a gating function. After employing a gating network with block-wise pooling and a customized FlashAttention kernel, the approach can achieve up to 5.67x speedups (for 32k contexts) while maintaining competitive perplexity scores.

**Strength:**

1. The idea of learning sparsity patterns using gating networks is straightforward yet effective.

2. It demonstrates strong speedups while maintaining long-context performance for up to 128k contexts.

3. It provides a customized FlashAttention kernel, enhancing both real-world applicability and efficiency.


**Weakness:**

1. Training cost concerns: Learning sparse attention requires additional training resources, making it less favorable compared to training-free sparse attention methods. Furthermore, a comprehensive comparison with both training-free sparse attention techniques and training-free context length extension methods is needed.

2. Limited novelty and lack of thorough analysis: The concept of learnable gating for pooling has been explored in prior works (e.g., TDSA), and adaptive sequence dimension compression has also been explored by methods such as QUEST and MInference. A more detailed analysis is necessary to clearly demonstrate the advantages of blockwise attention gating.

3. Clarity and writing issues: Certain methodological descriptions lack clarity, and there are issues with undefined or inconsistent symbols throughout the text.


**Reasons for the decision:**

While SeerAttention introduces an efficient mechanism for adaptive sparse attention and demonstrates promising results, it lacks a thorough analysis of its key advantages and contributions compared to training-free sparse attention methods and other sequence dimension compression techniques. Consequently, I am inclined to reject this paper.

**Additional Comments On Reviewer Discussion:**

During the rebuttal period, the common concerns raised by the reviewers and the corresponding author responses are provided as follows:

**1. Training cost and fair comparisons (6XFX, niUN):**

Reviewer Concerns: The reviewers raised concerns about the additional training cost introduced by SeerAttention and the lack of sufficient comparisons with training-free sparse attention methods plus context length extension methods.

Author Response: The authors clarified that the additional training cost is minimized through the use of a customized FlashAttention kernel and provided GPU-hour estimates alongside additional experiments to show that the proposed method can learn position information. However, a more thorough analysis is still needed to clearly demonstrate the advantages compared to training-free sparse attention methods and other sequence dimension compression techniques.

**2. Novelty and contributions (6XFX, niUN):**

Reviewer Concerns: Reviewers questioned the novelty of using learnable gating for sparse attention, citing similarities with prior methods such as TDSA and Quest. They requested further analysis to differentiate SeerAttention.

Author Response: The authors conducted ablations to highlight the advantages of blockwise attention gating and provided new comparisons with TDSA, demonstrating better performance.

**3. Evaluation on more downstream tasks (6XFX, 5GA1, niUN):**

Reviewer Concerns: The evaluation is performed on limited context length, with limited exploration of downstream benchmarks like Ruler and InfiniteBench.

Author Response: The authors added results from the LongBench and Ruler benchmarks, demonstrating improved performance and speedups over baselines.

**4. Clarity and writing (5GA1, niUN):**

Reviewer Concerns: Issues with unclear descriptions and inconsistent symbol usage were highlighted as barriers to understanding.

Author Response: The authors revised the manuscript to address these issues, improving clarity.
The rebuttal addressed some clarity and methodological concerns, and new experiments demonstrated SeerAttention's potential.

However, unresolved issues regarding novelty, the need for a more thorough analysis and comparison with related methods, and concerns about extra training costs have influenced the final decision. Consequently, I recommend rejection.

---

### Decision · Program_Chairs · 2025-01-22

Reject